# AlphaEdit$^+$: Model Editing in the Presence of Conflicting and Inconsistent Knowledge

## Abstract

Knowledge editing is a crucial technique for daily updates in LLMs, requiring a balance between accurately modifying incorrect knowledge and preserving existing information. The recently proposed AlphaEdit method achieves competitive editing performance by updating parameters under null-space constraints. However, our theoretical analysis reveals that AlphaEdit struggles with high knowledge conflicts and inconsistencies during editing. To address this, we propose a new editing method `AlphaEdit`$^+$, featuring three key improvements: 1) relaxing null-space constraints by adding a matrix perturbation through optimization to resolve conflicts between new and preserved knowledge; 2) introducing a weighting scheme on previously updated knowledge constraints to mitigate conflicts between new and historical editing; 3) developing a value smoothing algorithm to resolve high knowledge inconsistencies. These enhancements collectively ensure robust editing while maintaining model coherence. Comprehensive experiments show that our approach `AlphaEdit`$^+$ not only resolves the brittleness of the original method on carefully constructed challenging datasets but also achieves slightly better performance than AlphaEdit on existing benchmark datasets.

## 1 INTRODUCTION

Knowledge editing enables precise modifications to factual associations in LLMs, bypassing costly full retraining (Gupta et al., 2024; Zhang et al., 2024; Pan et al., 2025). Current approaches primarily comprise two paradigms: parameter-modifying techniques (e.g., UnKE (Deng et al., 2025), IFMET (Zhang et al., 2025), BaFT (Liu et al., 2025)) that directly edit critical model weights, and parameter-preserving methods (e.g., SERAC (Mitchell et al., 2022), MELO (Yu et al., 2024), postEdit (Song et al., 2024)) that store new knowledge in external modules or in-context editing via prompts(e.g., IKE (Zheng et al., 2023)). Although first-line techniques are efficient, the core challenge is integrating new knowledge while preserving existing capabilities, a balance difficult to achieve due to parameter shifting and limited effects on stored information. Recently, AlphaEdit (Fang et al., 2025) has emerged as an effective solution, strategically updating parameters via null-space projection to minimize interference with retained knowledge while ensuring accurate edits.

Despite significant advancements, existing knowledge editing studies critically overlook multifaceted conflicts between newly introduced knowledge, preserved foundational knowledge, and previously edited facts. In addition, severe inconsistencies may occur when edited knowledge diverges significantly from the model's existing parametric knowledge, manifested as logical contradictions (e.g., 'Paris is the capital of France' vs. new edit 'Roma is the capital of France') or cascading inference errors. Through mathematical analysis of AlphaEdit, we further characterize how these discrepancies destabilize its projection mechanism: highly conflicting between new edits and preserved knowledge spaces or pre-edits ultimately causing edit failure; and high levels of inconsistency exert significant adverse effects even on knowledge that is relatively easy to edit. While limited prior work has acknowledged these conflicts and inconsistencies, to the best of our knowledge, a systematic investigation and comprehensive solution remain unaddressed.

In this study, we formally quantify knowledge conflict and inconsistency for knowledge editing. We then propose `AlphaEdit`$^+$, a novel approach that systematically resolves these conflicts and inconsistency with three improvements. First, we introduce a perturbation matrix into the existing null-space matrix to alleviate its stringent constraints; this perturbation matrix is added as an additional

optimization variable within the overall objective function to resolve conflicts arising between new edits and existing knowledge. Additionally, we incorporate a conflict-based weighting factor for the residual terms of previous edits, reducing the negative effects of knowledge within previous edits that conflicts with new edits. Finally, for edits exhibiting high inconsistency, we adopt a progressive smoothing strategy for their objectives, facilitating incremental updates to the model parameters that progressively approaches the desired editing goal.

To investigate the limitations of existing approaches, we constructed three challenging datasets (AlphaSet1, AlphaSet2, and AlphaSet3) designed to target high-conflict and inconsistency scenarios. These are collectively referred to as AlphaSet. Experimental results reveal that existing methods suffer from a notable decline in editing scores under these settings, whereas our approach consistently demonstrates superior performance and robustness. These findings highlight that, in the presence of conflict and/or inconsistency, our method achieves progressively larger advantages over current SOTA techniques, particularly AlphaEdit. Our main contributions are summarized as follows:

- Building upon the AlphaEdit framework, we first establish a pioneering mathematical systematization of knowledge conflicts and inconsistencies in model editing, providing formal quantification and analyzing their detrimental impacts on editing performance.

- We then introduce `AlphaEdit`$^+$, a novel methodology incorporating three key enhancements: perturbation-based adaptive null-space matrix relaxation for conflict alleviation between new edits and model knowledge, conflict-aware weighting for regulating pre-edit constraints, and objective-oriented smoothing for progressively refining knowledge exhibiting inference inconsistencies.

- Through comprehensive comparative experiments accompanied by ablation studies and sensitive tests, we conclusively demonstrate the efficacy of our proposed approach.

## 2 THEORETICAL ANALYSES FOR ALPHAEDIT

### 2.1 PRELIMINARIES FOR ALPHAEDIT

Let $\mathbf{K}_0$, $\mathbf{K}_1$, and $\mathbf{K}_p$ be the key sets of the preserved, the to-be-updated, and previously updated knowledge, respectively. Their value sets are denoted by $\mathbf{V}_0$, $\mathbf{V}_1$, and $\mathbf{V}_p$, respectively. Let $\mathbf{W}$ be the parameters to be updated. Model editing seeks a perturbation $\mathbf{\Delta}$ of $\mathbf{W}$ so that a $(\mathbf{K}_1, \mathbf{V}_1)$ is correctly stored, while $(\mathbf{K}_0, \mathbf{V}_0)$ and $(\mathbf{K}_p, \mathbf{V}_p)$ are kept. Several classical methods, such as ROME (Meng et al., 2022) and AnyEdit (Jiang et al., 2025), have been developed based on this mechanism.

Recently, AlphaEdit (Fang et al., 2025) imposes a *null-space* constraint on $\mathbf{\Delta}$. Concretely, let $\mathbf{P}$ be the orthogonal projector onto the left null space of $\mathbf{K}_0$, constructed via the SVD of $\mathbf{K}_0\mathbf{K}_0^\top$ so that $\mathbf{P} = \hat{\mathbf{U}}\hat{\mathbf{U}}^\top$ with $\hat{\mathbf{U}}$ spanning the zero-eigen space. Consequently, the resulting objective[1] is:

$$\min_{\tilde{\mathbf{\Delta}}} \; \left\| (\mathbf{W} + \tilde{\mathbf{\Delta}}\mathbf{P})\mathbf{K}_1 - \mathbf{V}_1 \right\|_F^2 \; + \; \left\| \tilde{\mathbf{\Delta}}\mathbf{P}\mathbf{K}_p \right\|_F^2 \; + \; \lambda \| \tilde{\mathbf{\Delta}}\mathbf{P} \|_F^2, \tag{1}$$

where the second term penalizes interference with prior updated knowledge $\mathbf{K}_p$ (sequential editing) and the third term is a Tikhonov regularizer for numerical stability. Let $\mathbf{R} \triangleq \mathbf{V}_1 - \mathbf{W}\mathbf{K}_1$. Eq. 1 is a convex linear least-squares problem in $\mathbf{\Delta}$ and admits a closed form (Lang, 2012). The solution is:

$$\mathbf{\Delta}^* = \tilde{\mathbf{\Delta}}\mathbf{P} \; = \; \mathbf{R}\,\mathbf{K}_1^\top\mathbf{P} \left( \mathbf{K}_p\mathbf{K}_p^\top\mathbf{P} \; + \; \mathbf{K}_1\mathbf{K}_1^\top\mathbf{P} \; + \; \lambda\mathbf{I} \right)^{-1}, \tag{2}$$

where $\lambda$ is a hyperparameter. As $\mathbf{P}$ can be obtained in advance, computation is efficient. Experiments in Fang et al. (2025) sufficiently validate the superior performances of AlphaEdit.

### 2.2 DEFECT ANALYSIS OF ALPHAEDIT

This subsection mathematically demonstrates that AlphaEdit fails or produces suboptimal edits when severe knowledge conflicts and inconsistencies exist. We formally define knowledge conflict as follows: given a target edit $(\boldsymbol{k}, \boldsymbol{v})$, its conflict with a knowledge set $\mathbf{K}$ is quantified by:

$$s(\boldsymbol{k}, \mathbf{K}) \; = \; \max_{\boldsymbol{k}' \in \mathbf{K}} \frac{|\boldsymbol{k}^\top \boldsymbol{k}'|}{\|\boldsymbol{k}\| \, \|\boldsymbol{k}'\|} \; \in \; [0, 1]. \tag{3}$$

---

[1]In this study, we omit a minor weighting coefficient $\beta$ on the prior-edit term and this simplification does not affect the theoretical analysis (see Appendix A.5).

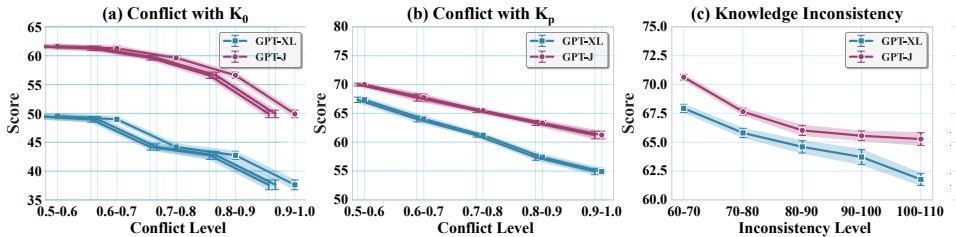

Figure 1: Edit scores versus conflict (similarity $s$) and inconsistency (residual $\|r\|$) for AlphaEdit on AlphaSet1, AlphaSet2, and AlphaSet3. The x-axis bins are left-open and right-closed intervals.

$\mathbf{K}$ can be $\mathbf{K}_0$ or $\mathbf{K}_p$. A high score reflects strong overlap with either preserved or previously updated knowledge, thereby indicates a high conflict between $k$ and $\mathbf{K}$. In addition, knowledge inconsistency is captured by the residual norm $\|r\| = \|v - \mathbf{W}k\|$, which measures the gap between the model's current knowledge (i.e., $\mathbf{W}k$) and the target value. A large $\|r\|$ implies a high inconsistency. In the following, we provide a detailed mathematical analysis of the impacts arising from three scenarios.

**Conflict with Preserved Knowledge ($\mathbf{K}_0$).** To simplify the analysis, we assume that $\mathbf{K}_1$ contains only a single knowledge tuple $(k_1, v_1)$ with $\|k_1\| = 1$ requiring editing, and we disregard $\mathbf{K}_p$. In this scenario, the AlphaEdit solution simplifies as $\mathbf{\Delta}^* = \mathbf{R}\, k_1^\top \mathbf{P} \left( k_1 k_1^\top \mathbf{P} + \lambda \mathbf{I} \right)^{-1}$. Through algebraic derivation, the solution simplifies to:

$$\mathbf{\Delta}^* = \frac{\lambda}{\lambda + \|\mathbf{P}k_1\|^2}\, r_1 \left( \mathbf{P}k_1 \right)^\top. \tag{4}$$

We first establish the relationship between this solution and the conflict coefficient $s$. The vector $k_1$ decomposes into components parallel and orthogonal to the column space of $\mathbf{K}_0$:

$$k_1 = k_{1\|} + k_{1\perp}, \quad \begin{cases} k_{1\|} \in \text{col}(\mathbf{K}_0) \\ k_{1\perp} \in \text{null}(\mathbf{K}_0^\top) \end{cases}. \tag{5}$$

After projection, $\mathbf{P}k_1 = k_{1\perp}$ (since $\mathbf{P}$ projects onto the nullspace). $s$ relates to the norms as:

$$\|k_{1\|}\|^2 \approx s^2, \quad \|k_{1\perp}\|^2 = \|k_1\|^2 - \|k_{1\|}\|^2 = 1 - s^2 \quad (\text{given } \|k_1\| = 1). \tag{6}$$

Therefore, $\|\mathbf{P}k_1\| = \|k_{1\perp}\| = \sqrt{1 - s^2}$. Then we have: $\mathbf{\Delta}^* = \frac{\lambda}{\lambda + 1 - s^2}\, r_1 \left( \mathbf{P}k_1 \right)^\top$. Accordingly, we have the following conclusion.

**Lemma 1** *The residual for $k_1$ after editing is:*

$$\|(\mathbf{W} + \mathbf{\Delta}^*)k_1 - v_1\| = \frac{\lambda}{\lambda + \|\mathbf{P}k_1\|^2}\, \|r_1\| = \frac{\lambda}{\lambda + 1 - s^2}\, \|r_1\|. \tag{7}$$

The proof of algebraic derivation and Lemma 1 is in the Appendix A.1. In the experiments, $\lambda \approx 0.1N$, where $N$ is the number of new edits. Therefore in this theoretical case, $\lambda$ can be considered as 0.1 (only one edit $(k_1, v_1)$). This lemma reveals that knowledge conflicts adversely affect editing success: smaller $s$ yields better edits (this conclusion well explains the observation of Duan et al. (2025), while $s \to 1$ results in complete edit failure (zero residual change). Our theoretical conclusion is further validated through real editing tasks. We construct three datasets(AlphaSet1, AlphaSet2 and AlphaSet3) to evaluate high conflicts with $\mathbf{K}_0$, high conflicts with $\mathbf{K}_p$, and high inconsistency, respectively. Using AlphaEdit, we analyze how editing performance relates to the degree of conflict or inconsistency; datasets details are in Section 4.1. As shown in Fig. 1(a), editing scores decrease consistently for both models as $\mathbf{K}_0$ conflict rises in AlphaSet1, with an average reduction of 11.82% from low to high conflict conditions.

**Conflict with Prior Edits ($\mathbf{K}_p$).** Consider a single new edit $(k_1, v_1)$ and one prior edit $k_p$ with $\|k_1\| = \|k_p\| = 1$. Assume both are orthogonal to the preserved knowledge ($\mathbf{K}_0^\top k_1 = \mathbf{K}_0^\top k_p = 0$), hence $\mathbf{P}k_1 = k_1$ and $\mathbf{P}k_p = k_p$. Therefore, $s = |k_1^\top k_p| \in [0, 1]$ and $\|r_1\| = \|v_1 - \mathbf{W}k_1\|$ denote the conflict and the inconsistency degrees, respectively. Restricting Eq. 1 to $\text{span}\{k_1, k_p\}$ and solving yields the closed-form update (detailed in the Appendix A.2):

$$\mathbf{\Delta}^* = r_1 k_1^\top \left( \lambda \mathbf{I} + k_1 k_1^\top + k_p k_p^\top \right)^{-1} = r_1 \frac{(\lambda + 1)\, k_1^\top - s\, k_p^\top}{(\lambda + 1)^2 - s^2}. \tag{8}$$

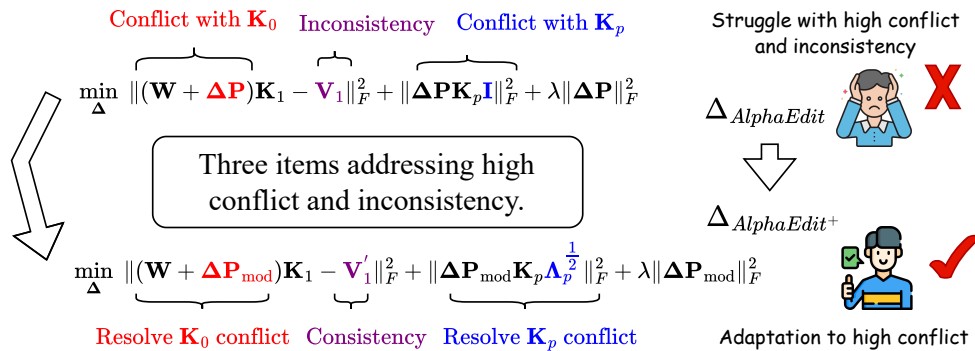

Figure 2: Comparison between AlphaEdit and our AlphaEdit$^+$. Best viewed in color.

Consequently, we obtain the following.

**Lemma 2** *The post-edit residual for the new edit satisfies:*

$$\left\|(\mathbf{W} + \boldsymbol{\Delta}^*)\boldsymbol{k}_1 - \boldsymbol{v}_1\right\| = \frac{\lambda(\lambda + 1)}{(\lambda + 1)^2 - s^2}\,\|\boldsymbol{r}_1\|. \tag{9}$$

When $s \to 0$ (no conflict with the prior edit), the factor reduces to $\frac{\lambda}{\lambda+1}$ (in this theoretical case, $\lambda$ can be considered as 0.2), recovering the single-edit case. As $|s| \to 1$, $(\lambda + 1)^2 - s^2$ shrinks and the residual increases, indicating stronger interference from prior edits. Fig. 1(b) shows on AlphaSet2, editing scores decrease by an average of 10.59% across both models as $\mathbf{K}_p$ conflict levels increase.

**Knowledge Inconsistency ($\mathbf{W}\boldsymbol{k}_1$ vs. $\boldsymbol{v}_1$).** We consider two edits to be updated:

$$\mathbf{K}_1 = [\boldsymbol{k}_{1,1}, \boldsymbol{k}_{1,2}], \quad \mathbf{V}_1 = [\boldsymbol{v}_{1,1}, \boldsymbol{v}_{1,2}], \quad \mathbf{R} = \mathbf{V}_1 - \mathbf{W}\mathbf{K}_1 = [\boldsymbol{r}_{1,1}, \boldsymbol{r}_{1,2}], \tag{10}$$

we disregard $\mathbf{K}_p$ and assume only the magnitude relation $\|\boldsymbol{r}_{1,2}\| = \rho\,\|\boldsymbol{r}_{1,1}\|$ with $\rho \geqslant 1$. Define the projection Gram matrix $\mathbf{G} \triangleq \mathbf{K}_1^\top \mathbf{P}\mathbf{K}_1$, which encodes the pairwise inner products of the projected keys and thereby collapses the high-dimensional problem onto $\mathrm{span}\{\mathbf{P}\boldsymbol{k}_{1,1}, \mathbf{P}\boldsymbol{k}_{1,2}\}$, enabling closed-form residuals via $(\mathbf{G} + \lambda\mathbf{I})^{-1}$.

$$\mathbf{G} \triangleq \mathbf{K}_1^\top \mathbf{P}\mathbf{K}_1 = \begin{bmatrix} g_{11} & g_{12} \\ g_{12} & g_{22} \end{bmatrix} \geqslant 0, \quad g_{ij} = \boldsymbol{k}_{1,i}^\top \mathbf{P}\boldsymbol{k}_{1,j}, \quad D_\lambda = (\lambda + g_{11})(\lambda + g_{22}) - g_{12}^2 > 0. \tag{11}$$

Solving the ridge objective restricted to the projected subspace gives the residual matrix identity:

$$\mathbf{E} \triangleq (\mathbf{W} + \boldsymbol{\Delta}^*)\mathbf{K}_1 - \mathbf{V}_1 = -\lambda\,\mathbf{R}\,(\mathbf{G} + \lambda\mathbf{I})^{-1}, \tag{12}$$

where

$$(\mathbf{G} + \lambda\mathbf{I})^{-1} = \frac{1}{D_\lambda}\begin{bmatrix} \lambda + g_{22} & -g_{12} \\ -g_{12} & \lambda + g_{11} \end{bmatrix}. \tag{13}$$

Let $\mathbf{E} = [\boldsymbol{e}_{1,1}, \boldsymbol{e}_{1,2}]$ denote the post-edit residual columns.

**Lemma 3** *If $\|\boldsymbol{r}_{1,2}\| = \rho\,\|\boldsymbol{r}_{1,1}\|$ with $\rho \geqslant 1$, the post-edit residual for the new edit satisfies:*

$$\|\boldsymbol{e}_{1,1}\| \leqslant \frac{\lambda}{D_\lambda}\Big((\lambda + g_{22}) + \rho\,|g_{12}|\Big)\|\boldsymbol{r}_{1,1}\|, \qquad \|\boldsymbol{e}_{1,2}\| \leqslant \frac{\lambda}{D_\lambda}\Big(|g_{12}| + \rho\,(\lambda + g_{11})\Big)\|\boldsymbol{r}_{1,1}\|. \tag{14}$$

The upper bound for the second key grows *linearly* with $\rho$ at slope $\frac{\lambda}{D_\lambda}(\lambda + g_{11})$, i.e., a larger initial residual on the second edit inevitably raises its post-edit residual bound. The first key's upper bound also increases linearly with $\rho$ at slope $\frac{\lambda}{D_\lambda}|g_{12}|$, reflecting cross-key coupling via $g_{12}$. Fig. 1(c) shows that on AlphaSet3, with increasing inconsistency levels, editing scores of both models decline, averaging a 5.76% reduction. Thus, editing algorithms must consider highly inconsistent data's harm.

## 3 METHODOLOGY

This section describes our proposed our new method AlphaEdit$^+$. First, we establish an iterative optimization objective to resolve the aforementioned knowledge conflicts and inconsistencies, along with an iterative solution framework and detailed algorithmic implementation.

## 3.1 A Unified Optimization Objective

As shown in Lemma 1, when the updated knowledge $\mathbf{K}_1$ is in high conflict with $\mathbf{K}_0$, the projected norm $\|\mathbf{P}\boldsymbol{k}_1\|$ approaches zero, leading to poor performance. To address this, one may either modify $\mathbf{K}_0$-which requires recomputing the SVD and incurs high computational cost-or adjust $\mathbf{K}_1$ in a semantically invariant way to reduce conflict. We propose a novel alternative by introducing a perturbation to $\mathbf{P}$. As indicated in Lemma 2, high conflict between $\mathbf{K}_1$ and $\mathbf{K}_p$ also severely impairs editing efficacy. To mitigate this, one can modify either $\mathbf{K}_1$ or $\mathbf{K}_p$ without altering their semantics to reduce conflict. This work prioritizes modifying $\mathbf{K}_1$ to implicitly reduce the influence of $\mathbf{K}_p$. Furthermore, Lemma 3 demonstrates that the presence of hard samples in the dataset can substantially affect overall performance. Motivated by curriculum learning, we adopt a progressive editing strategy that smooths such samples to dynamically adjust their difficulty. Building on these three insights, we formulate the following optimization objective:

$$\min_{\tilde{\boldsymbol{\Delta}}, \tilde{\boldsymbol{P}}} \left\| (\mathbf{W} + \tilde{\boldsymbol{\Delta}}\mathbf{P}_{\mathrm{mod}})\mathbf{K}_1 - \mathbf{V}_1^t \right\|_F^2 + \left\| \tilde{\boldsymbol{\Delta}}\mathbf{P}_{\mathrm{mod}}\mathbf{K}_p\boldsymbol{\Lambda}_p^{1/2} \right\|_F^2 + \lambda\left\| \tilde{\boldsymbol{\Delta}}\mathbf{P}_{\mathrm{mod}} \right\|_F^2, \tag{15}$$

$$\text{s.t.} \quad \mathbf{P}_{\mathrm{mod}} = \mathbf{P} + \tilde{\mathbf{P}}, \ \ \mathbf{P} \perp \tilde{\mathbf{P}}, \ \mathrm{rank}(\tilde{\mathbf{P}}) \text{ is minimized.}$$

where $\boldsymbol{\Lambda}_p = \mathrm{diag}(1 - |s(\boldsymbol{k}_j, \mathbf{K}_1)|)$, $\boldsymbol{k}_j \in \mathbf{K}_p$ and $\lambda$ is a hyperparameter; $\mathbf{V}_1^t$ denotes the smoothed targets[2] at iteration $t$:

$$\boldsymbol{v}_{1,i}^t = \boldsymbol{v}_{1,i} + \beta_i(t)\left(\boldsymbol{v}_{0,i} - \boldsymbol{v}_{1,i}\right), \qquad \beta_i(t) = \begin{cases} 0, & \text{if } \|\boldsymbol{r}_i\| \leqslant \tau_r, \\ \dfrac{T-t}{2T}, & \text{otherwise,} \end{cases} \tag{16}$$

with $\boldsymbol{r}_i = \boldsymbol{v}_{1,i} - \mathbf{W}\boldsymbol{k}_i$, current iteration $t$, threshold $\tau_r$, smoothing factor $\boldsymbol{\beta}$, and total iterations $T$. Eq. 15 yields our `AlphaEdit`$^+$. For AlphaEdit and AlphaEdit$^+$ compared in Fig. 2, we have:

**Lemma 4** *If* $\forall\boldsymbol{k}_i \in \mathbf{K}_1$, $s(\boldsymbol{k}_i, \mathbf{K}_0) \equiv 0$, $\|\boldsymbol{r}_i\| \leqslant \tau_r$, *and* $\forall\boldsymbol{k}_j \in \mathbf{K}_p, s(\boldsymbol{k}_j, \mathbf{K}_1) \equiv 0$, *then AlphaEdit$^+$ is reduced to AlphaEdit.*

The proof appears in Appendix A.4. The following section describe how to iteratively solve Eq. 15.

## 3.2 The Solving for Eq. (15)

Our solving relies on a validation set to select the optimal solution among the candidates generated by iteratively optimizing Eq. 15. Given that existing mainstream methods do not require validation sets, our validation set construction adheres to the following principles: 1) Fairness: The dataset must not leak any test set information to ensure unbiased method comparison; 2) Simplicity: The construction process should be straightforward, as excessive complexity hinders practical application. The specific construction steps include: 1) randomly sampling a subset of knowledge tuples for $\mathbf{K}_0$, $\mathbf{K}_1$, and $\mathbf{K}_p$, and 2) rewriting the selected samples via LLMs. Construction details for each experiment will be provided in the experimental section and Appendix B.1.

Since Eq. 15 contains two coupled optimization variables, we propose a two-stage optimization scheme to ensure effective solutions. In the first stage ($t = 0$), we optimize $\tilde{\mathbf{P}}$ independently as its introduction is unrelated to $\mathbf{V}_1$. Given $\mathbf{P}_{\mathrm{mod}} = \mathbf{P} + \tilde{\mathbf{P}}$, we have:

$$\boldsymbol{\Delta}_+^{(0)} = \tilde{\boldsymbol{\Delta}}\mathbf{P}_{\mathbf{mod}} = \mathbf{R}^0\mathbf{K}_1^\top\mathbf{P}_{\mathrm{mod}}\left(\mathbf{K}_p\boldsymbol{\Lambda}_p\mathbf{K}_p^\top\mathbf{P}_{\mathrm{mod}} + \mathbf{K}_1\mathbf{K}_1^\top\mathbf{P}_{\mathrm{mod}} + \lambda\mathbf{I}\right)^{-1}, \tag{17}$$

where $\mathbf{R}^0 = \mathbf{V}_1^0 - \mathbf{W}\mathbf{K}_1$. Assuming that $\mathbf{U}$ and $\mathbf{P}$ are given, let $\check{\mathbf{U}}$ be the set of sort eigenvectors with non-zero eigenvalues of $\mathbf{U}$ by ascending eigenvalue. We optimize $\tilde{\mathbf{P}}$ through a search-based approach, where at each search step we extract the eigenvector $\boldsymbol{u}_l$ corresponding to the current minimum eigenvalue from $\check{\mathbf{U}}$ (while removing it from $\check{\mathbf{U}}$ simultaneously), then update $\tilde{\mathbf{P}}$ as $\tilde{\mathbf{P}} = \tilde{\mathbf{P}} + \boldsymbol{u}_l\boldsymbol{u}_l^\top$. The modified $\mathbf{P}_{\mathrm{mod}}$ is substituted into Eq. 17 to compute the new $\tilde{\boldsymbol{\Delta}}$, followed by evaluating the current value of $\left\| (\mathbf{W} + \tilde{\boldsymbol{\Delta}}\mathbf{P}_{\mathrm{mod}})\mathbf{K}_1 - \mathbf{V}_1^0 \right\|_F^2 + \left\| \tilde{\boldsymbol{\Delta}}\mathbf{P}_{\mathrm{mod}}\mathbf{K}_p\boldsymbol{\Lambda}_p^{\frac{1}{2}} \right\|_F^2 + \lambda\left\| \tilde{\boldsymbol{\Delta}}\mathbf{P}_{\mathrm{mod}} \right\|_F^2$. If the total value is not reduced, then the search stops and the previous $\tilde{\mathbf{P}}$ is used as the solution.

---

[2]In this study, the average of $\boldsymbol{v}_1$ and $\boldsymbol{v}_0$ is set as the initial smoothed target.

---

**Algorithm 1:** AlphaEdit$^+$

---

**Input:** $\mathbf{K}_1, \mathbf{V}_1, \mathbf{K}_p, \mathbf{P}, \mathbf{U}$ and $\boldsymbol{\Sigma}$ from SVD($\mathbf{K}_0\mathbf{K}_0^\top$), $\mathbf{W}, T, \boldsymbol{\tau}_r, \epsilon, \delta, \lambda, \boldsymbol{\Lambda}_p = \mathbf{0}, \tilde{\mathbf{P}} = \mathbf{0}, \boldsymbol{\Delta}_+ = \mathbf{0}$,
**Output:** Final perturbation $\boldsymbol{\Delta}_+$
Update $\boldsymbol{\Lambda}_p$ with $\boldsymbol{\Lambda}_p[j,j] = 1 - |s(\boldsymbol{k}_j, \mathbf{K}_1)|, \ \boldsymbol{k}_j \in \mathbf{K}_p$;
Set $\breve{\mathbf{U}} \leftarrow \{\boldsymbol{u}_l : \ \sigma_l > 0, \sigma_l \in \boldsymbol{\Sigma}\}$ ordered by increasing $\sigma_l$;
Compute $\boldsymbol{R}^{(0)}, \mathbf{V}_1^{(0)}, \boldsymbol{\beta}(0)$, the objective in Eq. 15 with $t = 0$;
**for** *each $\boldsymbol{u}_l \in \breve{\mathbf{U}}$ in order* **do**
    $\tilde{\mathbf{P}} \leftarrow \tilde{\mathbf{P}} + \boldsymbol{u}_l\boldsymbol{u}_l^\top$;
    Recompute the objective in Eq. 15 with $t = 0$;
    **if** *value reduction $< \epsilon$* **then** $\mathbf{P}_{\text{mod}}^* \leftarrow \mathbf{P} + \tilde{\mathbf{P}} - \boldsymbol{u}_l\boldsymbol{u}_l^\top$; break;
**if** $\|\boldsymbol{\beta}(0)\|_1 = 0$ **then** $\boldsymbol{\Delta}_+ \leftarrow$ Eq. 17 using $\mathbf{P}_{\text{mod}}^*$; **return** $\boldsymbol{\Delta}_+$ ;
Compute $\boldsymbol{\Delta}_+^{(0)}$ with Eq. 17; evaluate val_score$^{(0)}$;
**for** $t = 1$ **to** $T$ **do**
    Compute $\boldsymbol{R}^{(t)}, \mathbf{V}_1^{(t)}, \boldsymbol{\beta}(t)$;
    Compute $\boldsymbol{\Delta}_+^{(t)}$ with Eq. 18; evaluate val_score$^{(t)}$;
    **if** $\left|val\_score^{(t)} - val\_score^{(t-1)}\right| \leqslant \delta$ **then** break;
**return** $\boldsymbol{\Delta}_+^{(t)}$;

---

In the second stage ($t \geqslant 1$), we fixed $\mathbf{P}_{\text{mod}}^*$ and search the optimal $\tilde{\boldsymbol{\Delta}}$. In the $t$-th iteration, the current temporal optimal solution is as follows:

$$\boldsymbol{\Delta}_+^{(t)} = \boldsymbol{R}^t\mathbf{K}_1^\top\mathbf{P}_{\text{mod}}^* \left(\mathbf{K}_p\boldsymbol{\Lambda}_p\mathbf{K}_p^\top\mathbf{P}_{\text{mod}}^* + \mathbf{K}_1\mathbf{K}_1^\top\mathbf{P}_{\text{mod}}^* + \lambda\mathbf{I}\right)^{-1}, \tag{18}$$

where $\mathbf{R}^t = \mathbf{V}_1^t - \mathbf{W}\mathbf{K}_1$ and $\boldsymbol{\Delta}_+^{(t)} = \tilde{\boldsymbol{\Delta}}\mathbf{P}_{\text{mod}}^*$. For this temporary optimal solution, we evaluate it using the previously constructed validation set and record the performance score. The optimization stops when the performance score no longer increases or when $t > T$.

The steps of the entire solving procedure is presented in Algorithm 1. Compared to AlphaEdit, the overall computational overhead primarily stems from repeated evaluations of Eqs. 17 and 18. Since only the term involving $\mathbf{R}^t$ changes in Eq. 18, other components can be reused to accelerate the process. Overall, the additional computational cost remains moderate, as confirmed experimentally, with only a marginal increase observed. Future work will explore efficient approximations for inverting the matrix in Eq. 17 to further enhance computational efficiency.

# 4 EXPERIMENTS

We conduct systematic experiments to answer three core questions: (1) robustness & efficacy: whether AlphaEdit$^+$ reduces editing failures under high conflict with preserved knowledge and prior edits, as well as high inconsistency, and its runtime overhead relative to AlphaEdit; (2) general capability preservation: how well models edited by AlphaEdit$^+$ retain downstream abilities after sequences of challenging edits; and (3) effects on hidden representations: whether the new components (projection perturbation, conflict-aware weighting, value smoothing) alter representations of unedited knowledge.

## 4.1 EXPERIMENT SETUP

**Base LLMs & Baselines.** Our experiments are conducted on two LLMs: GPT2-XL (1.5B) (Radford et al., 2019) and GPT-J (6B) (Wang & Komatsuzaki, 2021). We compare our method against several model editing baselines, including MEMIT (Meng et al., 2023), PRUNE (Ma et al., 2025), RECT (Gu et al., 2024), and AlphaEdit (Fang et al., 2025). For all baselines, we use their original settings with implementations from the AlphaEdit repository. Unless otherwise noted, we use the following hyperparameters throughout: shared settings $T = 10$, $\lambda = 10$ and $\delta = 2 \times 10^{-4}$ for both models; and model-specific settings $\epsilon = 3.0 \times 10^{-4}$ for GPT-2-XL and $1.2 \times 10^{-4}$ for GPT-J (Algorithm 1). All experiments were conducted on a single NVIDIA A100 GPU (80 GB).

**Datasets and Metrics.** As noted in Section 2.2, we construct three datasets. First, we introduce a new dataset, **AlphaSet**, built upon the widely used model-editing benchmarks ZsRE (Levy et al.,

Table 1: Mean conflict/inconsistency levels and overall results on five datasets. Best per block in **bold**.

| Dataset | Method | GPT2-XL (1.5B) | | | | GPT-J (6B) | | | |
|---|---|---|---|---|---|---|---|---|---|
| | | Eff↑ | Gen↑ | Spe↑ | Score↑ | Eff↑ | Gen↑ | Spe↑ | Score↑ |
| **AlphaSet1** | MEMIT | 70.66 | 56.44 | 22.22 | 49.77 | 98.95 | 90.19 | 27.10 | 72.08 |
| $\text{avg}(s_{K_0}) = 0.79$ | RECT | 49.76 | 42.57 | 22.35 | 38.23 | 87.98 | 69.05 | 26.80 | 61.28 |
| $\text{avg}(s_{K_p}) = 0.38$ | PRUNE | 92.50 | 82.54 | 23.77 | 66.27 | 96.91 | 91.13 | 27.76 | 71.93 |
| $\text{avg}(\|r\|) = 17.67$ | AlphaEdit | 78.67 | 63.45 | 22.64 | 54.92 | 97.24 | 82.33 | 27.15 | 68.91 |
| | **AlphaEdit$^+$** | **97.40** | **87.48** | **23.89** | **69.59** | **99.74** | **91.33** | **27.82** | **72.96** |
| **AlphaSet2** | MEMIT | 75.78 | 63.40 | 24.18 | 54.45 | 85.19 | 78.31 | **27.43** | 63.64 |
| $\text{avg}(s_{K_0}) = 0.42$ | RECT | 65.91 | 52.12 | 24.20 | 47.41 | 82.08 | 68.60 | 27.13 | 59.27 |
| $\text{avg}(s_{K_p}) = 0.81$ | PRUNE | 78.62 | 70.76 | 24.26 | 57.88 | 78.25 | 71.83 | 27.34 | 59.14 |
| $\text{avg}(\|r\|) = 12.54$ | AlphaEdit | 88.27 | 75.87 | 24.38 | 62.84 | 96.02 | **79.93** | 26.99 | 67.65 |
| | **AlphaEdit$^+$** | **95.31** | **76.59** | **24.42** | **65.44** | **98.91** | 79.19 | 27.01 | **68.44** |
| **AlphaSet3** | MEMIT | 58.38 | 47.28 | 22.41 | 42.69 | 93.53 | 83.87 | 25.72 | 67.71 |
| $\text{avg}(s_{K_0}) = 0.39$ | RECT | 39.53 | 32.62 | 21.24 | 31.13 | 81.17 | 61.98 | 25.41 | 56.19 |
| $\text{avg}(s_{K_p}) = 0.36$ | PRUNE | 80.12 | 72.81 | **22.42** | 58.45 | 93.53 | 83.87 | 25.75 | 67.72 |
| $\text{avg}(\|r\|) = 57.29$ | AlphaEdit | 79.10 | 74.58 | 20.31 | 58.00 | 92.94 | 84.65 | 24.84 | 67.48 |
| | **AlphaEdit$^+$** | **80.13** | **75.33** | 20.31 | **58.59** | **93.94** | **85.41** | **25.97** | **68.21** |
| **ZsRE** | MEMIT | 87.22 | 83.21 | 26.41 | 65.61 | 98.97 | 98.54 | 27.73 | 75.08 |
| $\text{avg}(s_{K_0}) = 0.53$ | RECT | 77.90 | 69.85 | 25.60 | 57.78 | 96.95 | 93.07 | 28.04 | 72.69 |
| $\text{avg}(s_{K_p}) = 0.62$ | PRUNE | 84.85 | 82.09 | 27.47 | 64.80 | 96.04 | 95.84 | **34.17** | 75.35 |
| $\text{avg}(\|r\|) = 21.6$ | AlphaEdit | 97.85 | 93.26 | 26.56 | 72.56 | 99.66 | **99.15** | 27.70 | 75.50 |
| | **AlphaEdit$^+$** | **98.81** | **94.32** | **27.73** | **73.62** | **99.76** | 99.09 | 32.42 | **77.09** |
| **Counterfact** | MEMIT | 87.00 | 48.75 | **12.95** | 49.57 | 92.50 | 67.50 | 14.35 | 58.11 |
| $\text{avg}(s_{K_0}) = 0.23$ | RECT | 67.00 | 31.75 | 12.00 | 36.92 | 96.00 | 48.75 | 14.50 | 53.08 |
| $\text{avg}(s_{K_p}) = 0.47$ | PRUNE | 90.00 | 69.25 | 10.40 | 56.55 | 93.32 | **72.50** | 12.65 | 59.49 |
| $\text{avg}(\|r\|) = 88.54$ | AlphaEdit | 95.50 | 66.75 | 10.75 | 57.67 | 97.50 | 70.75 | 14.15 | 60.80 |
| | **AlphaEdit$^+$** | **96.63** | **69.50** | 11.45 | **59.19** | **98.78** | 71.00 | **14.75** | **61.51** |

2017) and Counterfact (Meng et al., 2022), and further augmented with samples derived from Wikipedia (Vrandečić & Krötzsch, 2014). AlphaSet consists of three subsets: conflict with preserved knowledge (AlphaSet1), conflict with prior edits (AlphaSet2), and knowledge inconsistency (AlphaSet3), comprising a total of 3,500 examples. For experiments involving smoothing, we design dedicated validation sets to identify the overall optimal solution (Appendix B.1). Second, we retain the original ZsRE and Counterfact benchmarks as separate testbeds to verify effectiveness on standard datasets. Third, to assess downstream general abilities, we use six tasks: SST (Socher et al., 2013), MRPC (Dolan & Brockett, 2005), MMLU (Hendrycks et al., 2021), RTE (Bentivogli et al., 2009), CoLA (Warstadt et al., 2019), and NLI (Williams et al., 2018). To evaluate editing performance, we adopt the classical metrics of Efficacy (edit success), Generalization (paraphrase success), and Specificity (neighborhood preservation), together with their macro-average Score, where higher values indicate superior performance. Additionally, we report F1 scores on six downstream tasks to assess the retention of general capabilities.

## 4.2 Performance on five datesets

Prior to experimentation, we computed and analyzed the distributions of two conflict types and one inconsistency across all five datasets, each comprising 1,000 instances. As shown in Fig. 3, $\mathbf{K}_0$ conflicts are most pronounced in AlphaSet1, while $\mathbf{K}_p$ conflicts are most severe in AlphaSet2, compared to other datasets. Inconsistencies are extremely severe in Counterfact and second most severe in AlphaSet3. These observations confirm that our constructed datasets well reflect knowledge conflict and inconsistency characteristics. To evaluate the core efficacy and robustness of AlphaEdit$^+$, we experiment on the above three challenging datasets and further test it on two standard benchmarks, ZsRE and Counterfact. Since our algorithm addresses inconsistent knowledge within samples rather than targeting globally large residuals, we set the inconsistency threshold to $\tau_r = 100$ for the Counterfact dataset and to $\tau_r = 30$ for the rest (results with more thresholds are in Appendix C.4).

As shown in Table 1, AlphaEdit$^+$ consistently outperforms AlphaEdit and other baselines across nearly all metrics. Efficacy and Generalization rise by 5.53% and 4.58% on average across two models, while Specificity is preserved or even improved. On the existing datasets, AlphaEdit$^+$ performs competitively: on ZsRE, it achieves the highest Efficacy and overall editing score; on Counterfact, AlphaEdit$^+$ delivers slight improvements across all metrics compared to AlphaEdit. These gains can be attributed to the moderate levels of conflict and inconsistency observed in both

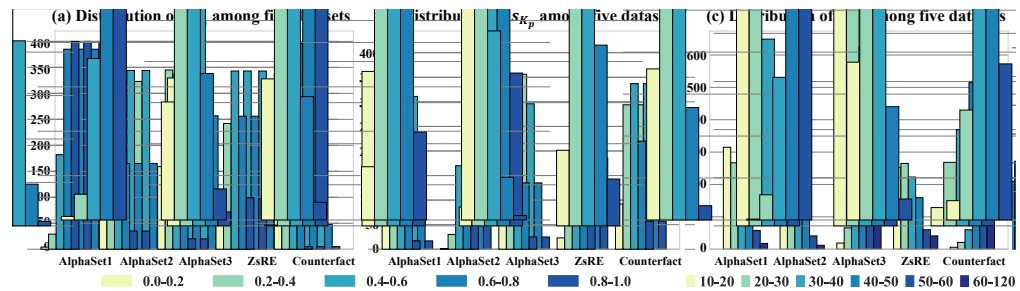

Figure 3: Distributions of $\mathbf{K}_0$ conflict, $\mathbf{K}_p$ conflict, and inconsistency across five datasets.

datasets. Overall, AlphaEdit$^+$ improves the average editing score by 3.90% over AlphaEdit on our constructed datasets and achieves an additional 1.22% average improvement on standard benchmarks.

As shown in Table 2, our method requires three times the runtime of AlphaEdit on AlphaSet3, mainly from smoothing iterations. However, absolute editing times remain low, making the overhead acceptable given the performance gains. Future work will investigate acceleration strategies, including distributed and parallel solvers, to reduce overhead and improve scalability.

To further assess the intrinsic knowledge of post-edited LLMs, we conduct General Capability Tests on six tasks from the GLUE benchmark (Wang et al., 2018). Specifically, we evaluate AlphaEdit$^+$ on AlphaSet, ZsRE, and Counterfact to compare the general capabilities of two models before and after editing. The evaluation, summarized in Fig. 4, is performed after sequentially applying 2,000 edits. The results indicate minimal impact on the models' general capabilities: except for a marginal decline of 0.23 on SST-2 in GPT2-XL, original capabilities remain well preserved across all settings. Overall, these results confirm that the three modules preserve general capabilities, even under large-scale sequential editing.

Table 2: Runtime of AlphaEdit and AlphaEdit$^+$ on GPT2-XL.

| Dataset | AlphaEdit | AlphaEdit$^+$ |
|---|---|---|
| AlphaSet1 | 112.69s | 170.60s |
| AlphaSet2 | 106.46s | 100.34s |
| AlphaSet3 | 199.05s | 685.68s |
| ZsRE | 490.25s | 614.44s |
| Counterfact | 476.79s | 675.45s |

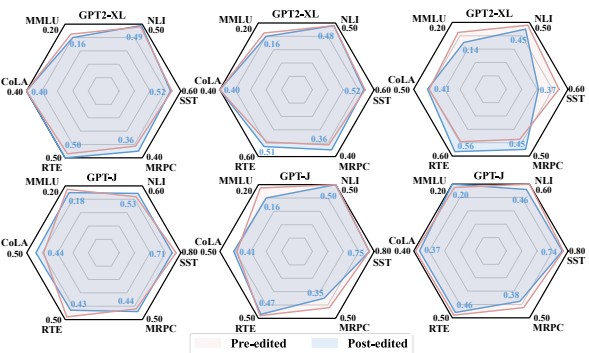

Figure 4: The impact of AlphaEdit$^+$ on the general F1 scores of two models, evaluated on the AlphaSet, ZsRE, and CounterFact datasets. Best viewed in color.

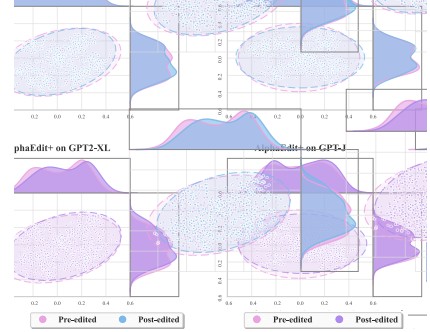

Figure 5: The distribution of hidden representations after dimensionality reduction.

### 4.3 HIDDEN STATE ANALYSIS AND ABLATION EXPERIMENTS

For analysis the hidden state we randomly select 1,000 factual prompts and extract the hidden representations within pre-edited LLMs. Subsequently, we performed AlphaSet, ZsRE and Counterfact on the LLMs and recomputed these hidden representation. Finally, we used t-SNE (Maaten & Hinton, 2008) to visualize the hidden representation before and after editing. Fig. 5 exhibits them and their marginal distribution curves. AlphaEdit$^+$ preserves consistency in hidden representations with AlphaEdit after editing. Specifically, in LLMs edited by AlphaEdit$^+$, the hidden representations remain aligned with the original distribution across both base

Table 3: Comprehensive ablation analysis of AlphaEdit$^+$ on AlphaSet: optimal results per metric highlighted in **Bold**.

| Variant | GPT2-XL (1.5B) | | | | GPT-J (6B) | | | |
|---|---|---|---|---|---|---|---|---|
| | Eff↑ | Gen↑ | Spe↑ | Score↑ | Eff↑ | Gen↑ | Spe↑ | Score↑ |
| AlphaEdit$^+$ | **93.33** | **82.69** | **26.21** | **67.41** | **96.25** | **87.84** | **29.50** | **71.20** |
| w/o $\tilde{\mathbf{P}}$ | 92.60 | 82.61 | 25.85 | 67.02 | 95.75 | 87.42 | 29.29 | 70.82 |
| w/o $\mathbf{\Lambda}_p$ | 93.31 | 81.90 | 26.19 | 67.13 | 95.62 | 86.96 | 28.94 | 70.51 |
| w/o Smoothing | 93.14 | 80.43 | 25.77 | 66.45 | 95.25 | 87.15 | 29.39 | 70.60 |

models. This stability indicates that the three components introduced in AlphaEdit$^+$—projection perturbation, conflict-aware weighting, and target smoothing—do not induce distributional shifts in the model's hidden representations, while simultaneously mitigating overfitting.

Beyond the overall improvements, we further conduct an ablation study to disentangle the contribution of each component. Table 3 reports ablations on AlphaSet, which includes conflicts and inconsistencies. Disabling each module leads to measurable drops: removing projection perturbation slightly weakens efficacy and specificity, removing conflict-aware weighting reduces generalization, and discarding smoothing causes the largest decline in both generalization and overall score. These results confirm that the three components play complementary roles, and that the full AlphaEdit$^+$ consistently achieves the most balanced performance across both models.

## 5 RELATED WORK

**Knowledge Editing.** Parameter-modifying knowledge editing methods enable efficient factual updates LLMs via direct weight adjustments, avoiding costly full retraining. Early approaches like ROME (Meng et al., 2022) (rank-one weight updates for single facts) and MEMIT (Meng et al., 2023) (multi-layer updates for scaling) prioritize new edit accuracy but lack constraints, risking interference with existing knowledge. Later, AlphaEdit (Fang et al., 2025) introduced null-space projection to confine updates to the orthogonal subspace of preserved knowledge, boosting pre-stored information preservation. Complementary efforts include AnyEdit (Jiang et al., 2025) and SIR (Wang et al., 2025a) (enhancing edit generalization/efficiency) and PRUNE (Ma et al., 2025) (numerical restraints for retaining general capabilities). Yet these methods rarely place systematic focus on the complex relationships that may exist between new, historical, and preserved knowledge, as well as their impact on editing performance—and this creates room for our research.

**Learning Under Imperfect Data.** Models are often trained or updated under imperfect supervision, such as noisy labels (Song et al., 2022) or intrinsically hard examples (Zhou et al., 2025), which has motivated two classic strategies: example weighting (Xie et al., 2025) and target/data smoothing (Rangwani et al., 2022). Weighting methods down-weight harmful signals; meta-reweighting learns per-example weights from a small clean set to improve robustness (Ren et al., 2018). Sequential smoothing and differencing denoise and stabilize time-series data, substantially improving deep-learning forecasting (Livieris et al., 2021). In knowledge editing, recent studies have begun to investigate edit difficulty. SCIENCEMETER (Wang et al., 2025b) reports scientific frontier/ambiguous claims are prone to erroneous updates; SGR-Edit (Chen et al., 2025) shows short-answer edits tend to overfit, while evidence-based rationales generalize better. Ge et al. (2024) used target "perplexingness" to measure edit difficulty. Yet perplexity is static. This studies introduce a dynamic, optimization-aligned difficulty indicator, namely, the editing residual, which is more consistence with previous deep learning studies that use training losses as difficulty indicator.

## 6 CONCLUSION

This work tackles an underexplored failure mode in knowledge editing—edits that conflict with preserved/prior knowledge or diverge from a model's parametric beliefs—and shows, theoretically and empirically, that AlphaEdit's null-space projection degrades in these regimes. We present AlphaEdit$^+$, which addresses this via three components: (1) a learnable projection perturbation to relax rigid constraints, (2) conflict-aware weighting to reduce interference from prior edits, and (3) progressive target smoothing for large-residual edits. Across our challenging dataset AlphaSet and standard benchmarks, on GPT2-XL and GPT-J, AlphaEdit$^+$ increases edit success and paraphrase generalization while maintaining (or modestly improving) specificity and preserving general capabilities, with only moderate overhead. These results indicate that principled control of projection geometry, edit weighting, and objective smoothing enables robust, minimally disruptive editing. In future work, we will investigate better edit difficulty measures, refine the smoothing design to better handle high inconsistency, improve efficiency and outcomes, and explore transferring our approach to other parameter-modifying model-editing frameworks.

ETHICS STATEMENT

This work aims to advance knowledge editing techniques in large language models (LLMs). Our proposed framework, AlphaEdit$^{+}$, is designed to improve robustness in the presence of conflicting and inconsistent knowledge without requiring full retraining. We emphasize that this research is intended solely for responsible scientific exploration, supporting safe, transparent, and efficient model maintenance. We do not attempt to use model editing techniques for generating harmful, biased, or misleading content. The datasets employed (ZsRE, Counterfact, and Wikipedia) are all publicly available, widely adopted in prior research, and contain no personally identifiable or sensitive information. The methods and results presented in this paper are intended to promote reproducibility and enable critical evaluation by the research community, in line with ethical standards for AI research.

REPRODUCIBILITY STATEMENT

We are committed to ensuring the reproducibility of our research findings. To this end, we provide comprehensive details of our experimental setup, dataset construction, hyperparameter configurations, and evaluation procedures in Section 4.1 and Appendix B. For the review process, we have released our complete codebase together with selected portions of the constructed datasets in an anonymized repository (Anonymized repository link: `https://anonymous.4open.science/r/AlphaEdit_plus-B5F8`). In addition, a compressed version of the code is included in the Supplementary Material. Upon acceptance, we will make the entire codebase and all datasets publicly available through a permanent GitHub repository.

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

USE OF LARGE LANGUAGE MODELS

In accordance with ICLR policy, we disclose that LLMs were used only for grammar and style polishing. All ideas and analyses are by the authors, who take full responsibility for the content.

## A  THEORETICAL ANALYSIS

This section provides detailed mathematical derivations for the results and lemmas in Section 2. We strictly follow the notations used in the main text.

### A.1  CONFLICT WITH PRESERVED KNOWLEDGE ($\mathbf{K}_0$)

We consider the simplified case with a single edit $(\boldsymbol{k}_1, \boldsymbol{v}_1)$ while ignoring $\mathbf{K}_p$. In this situation, the AlphaEdit solution reduces from Eq. 2 to

$$\boldsymbol{\Delta}^* = \mathbf{R}\, \boldsymbol{k}_1^\top \mathbf{P} \left( \boldsymbol{k}_1 \boldsymbol{k}_1^\top \mathbf{P} + \lambda \mathbf{I} \right)^{-1}, \qquad \mathbf{R} = \boldsymbol{r}_1. \tag{19}$$

Since $\boldsymbol{k}_1 \boldsymbol{k}_1^\top \mathbf{P}$ has rank at most one, let us denote $\boldsymbol{u} = \mathbf{P}\boldsymbol{k}_1$. Then $\boldsymbol{k}_1 \boldsymbol{k}_1^\top \mathbf{P} = \boldsymbol{k}_1 \boldsymbol{u}^\top$, and the inverse becomes $(\lambda \mathbf{I} + \boldsymbol{k}_1 \boldsymbol{u}^\top)^{-1}$. By the Sherman–Morrison formula,

$$(\lambda \mathbf{I} + \boldsymbol{k}_1 \boldsymbol{u}^\top)^{-1} = \frac{1}{\lambda}\mathbf{I} - \frac{1}{\lambda^2} \frac{\boldsymbol{k}_1 \boldsymbol{u}^\top}{1 + \frac{1}{\lambda}\boldsymbol{u}^\top \boldsymbol{k}_1}. \tag{20}$$

Substituting this expression back into the solution yields

$$\boldsymbol{\Delta}^* = \boldsymbol{r}_1\, \boldsymbol{k}_1^\top \mathbf{P} \left( \frac{1}{\lambda}\mathbf{I} - \frac{1}{\lambda^2} \frac{\boldsymbol{k}_1 \boldsymbol{u}^\top}{1 + \frac{1}{\lambda}\boldsymbol{u}^\top \boldsymbol{k}_1} \right). \tag{21}$$

Since $\boldsymbol{k}_1^\top \mathbf{P} = \boldsymbol{u}^\top$, the first term becomes $\frac{1}{\lambda}\,\boldsymbol{r}_1 \boldsymbol{u}^\top$, while the second term equals

$$-\boldsymbol{r}_1 \frac{1}{\lambda^2} \cdot \frac{\boldsymbol{u}^\top \boldsymbol{k}_1}{1 + \frac{1}{\lambda}\boldsymbol{u}^\top \boldsymbol{k}_1}\, \boldsymbol{u}^\top. \tag{22}$$

Factoring out $\boldsymbol{r}_1 \boldsymbol{u}^\top$, the coefficient is

$$\frac{1}{\lambda} - \frac{1}{\lambda^2} \cdot \frac{\boldsymbol{u}^\top \boldsymbol{k}_1}{1 + \frac{1}{\lambda}\boldsymbol{u}^\top \boldsymbol{k}_1}. \tag{23}$$

Since $\boldsymbol{u} = \mathbf{P}\boldsymbol{k}_1$ and $\mathbf{P}$ is a projection, we have $\boldsymbol{u}^\top \boldsymbol{k}_1 = \|\boldsymbol{u}\|^2$. Denoting $\|\boldsymbol{u}\|^2 = \|\mathbf{P}\boldsymbol{k}_1\|^2$, the coefficient simplifies to $\frac{1}{\lambda + \|\mathbf{P}\boldsymbol{k}_1\|^2}$. Hence the closed-form solution becomes

$$\boldsymbol{\Delta}^* = \frac{1}{\lambda + \|\mathbf{P}\boldsymbol{k}_1\|^2}\, \boldsymbol{r}_1\, \boldsymbol{u}^\top = \frac{\lambda}{\lambda + \|\mathbf{P}\boldsymbol{k}_1\|^2}\, \boldsymbol{r}_1\, (\mathbf{P}\boldsymbol{k}_1)^\top, \tag{24}$$

which matches the expression in the main text.

Finally, the post-edit residual is

$$(\mathbf{W} + \boldsymbol{\Delta}^*)\boldsymbol{k}_1 - \boldsymbol{v}_1 = -\frac{\lambda}{\lambda + \|\mathbf{P}\boldsymbol{k}_1\|^2}\, \boldsymbol{r}_1, \tag{25}$$

and therefore

$$\|(\mathbf{W} + \boldsymbol{\Delta}^*)\boldsymbol{k}_1 - \boldsymbol{v}_1\| = \frac{\lambda}{\lambda + \|\mathbf{P}\boldsymbol{k}_1\|^2}\, \|\boldsymbol{r}_1\|. \tag{26}$$

### A.2  CONFLICT WITH PRIOR EDITS ($\mathbf{K}_p$)

We consider a new edit $(\boldsymbol{k}_1, \boldsymbol{v}_1)$ and one prior edit $\boldsymbol{k}_p$, both assumed to have unit norm and to be orthogonal to $\mathbf{K}_0$. In this case $\mathbf{P}\boldsymbol{k}_1 = \boldsymbol{k}_1$ and $\mathbf{P}\boldsymbol{k}_p = \boldsymbol{k}_p$, so the objective reduces to

$$\min_{\boldsymbol{\Delta}} \|\boldsymbol{\Delta}\boldsymbol{k}_1 - \boldsymbol{r}_1\|^2 + \|\boldsymbol{\Delta}\boldsymbol{k}_p\|^2 + \lambda\|\boldsymbol{\Delta}\|_F^2. \tag{27}$$

Since the optimization only involves the directions $\boldsymbol{k}_1$ and $\boldsymbol{k}_p$, the optimal $\boldsymbol{\Delta}$ necessarily lies in the row space spanned by $\boldsymbol{k}_1^\top$ and $\boldsymbol{k}_p^\top$. From Eq. (2), the solution can be written as

$$\boldsymbol{\Delta}^* = \boldsymbol{r}_1 \boldsymbol{k}_1^\top \big(\lambda \mathbf{I} + \boldsymbol{k}_1 \boldsymbol{k}_1^\top + \boldsymbol{k}_p \boldsymbol{k}_p^\top\big)^{-1}. \tag{28}$$

To evaluate this expression, we restrict attention to the two-dimensional subspace $\operatorname{span}\{\boldsymbol{k}_1, \boldsymbol{k}_p\}$. The Gram matrix in this subspace is

$$\mathbf{G} = \begin{bmatrix} 1 & s \\ s & 1 \end{bmatrix}, \qquad s = \boldsymbol{k}_1^\top \boldsymbol{k}_p. \tag{29}$$

Therefore,

$$\lambda \mathbf{I} + \mathbf{G} = \begin{bmatrix} \lambda + 1 & s \\ s & \lambda + 1 \end{bmatrix}, \quad (\lambda \mathbf{I} + \mathbf{G})^{-1} = \frac{1}{(\lambda+1)^2 - s^2} \begin{bmatrix} \lambda + 1 & -s \\ -s & \lambda + 1 \end{bmatrix}. \tag{30}$$

In this coordinate system, $\boldsymbol{k}_1^\top$ corresponds to $[1, 0]$, and hence

$$\boldsymbol{k}_1^\top (\lambda \mathbf{I} + \mathbf{G})^{-1} = \frac{1}{(\lambda+1)^2 - s^2} \big((\lambda+1)\boldsymbol{k}_1^\top - s\boldsymbol{k}_p^\top\big). \tag{31}$$

Substituting this back, the closed-form update becomes

$$\boldsymbol{\Delta}^* = \boldsymbol{r}_1 \frac{(\lambda+1)\boldsymbol{k}_1^\top - s\boldsymbol{k}_p^\top}{(\lambda+1)^2 - s^2}. \tag{32}$$

Finally, multiplying by $\boldsymbol{k}_1$ yields the residual

$$(\mathbf{W} + \boldsymbol{\Delta}^*)\boldsymbol{k}_1 - \boldsymbol{v}_1 = -\frac{\lambda(\lambda+1)}{(\lambda+1)^2 - s^2} \boldsymbol{r}_1, \tag{33}$$

and therefore

$$\big\|(\mathbf{W} + \boldsymbol{\Delta}^*)\boldsymbol{k}_1 - \boldsymbol{v}_1\big\| = \frac{\lambda(\lambda+1)}{(\lambda+1)^2 - s^2} \|\boldsymbol{r}_1\|. \tag{34}$$

### A.3 Knowledge Inconsistency

We now consider two new edits

$$\mathbf{K}_1 = [\boldsymbol{k}_{1,1}, \boldsymbol{k}_{1,2}], \quad \mathbf{V}_1 = [\boldsymbol{v}_{1,1}, \boldsymbol{v}_{1,2}], \quad \mathbf{R} = [\boldsymbol{r}_{1,1}, \boldsymbol{r}_{1,2}], \tag{35}$$

with $\|\boldsymbol{r}_{1,2}\| = \rho\|\boldsymbol{r}_{1,1}\|$. Define the projected Gram matrix

$$\mathbf{G} = \mathbf{K}_1^\top \mathbf{P} \mathbf{K}_1 = \begin{bmatrix} g_{11} & g_{12} \\ g_{12} & g_{22} \end{bmatrix}, \qquad D_\lambda = (\lambda + g_{11})(\lambda + g_{22}) - g_{12}^2 > 0. \tag{36}$$

From the ridge regression form, the post-edit residual matrix satisfies

$$\mathbf{E} \triangleq (\mathbf{W} + \boldsymbol{\Delta}^*)\mathbf{K}_1 - \mathbf{V}_1 = -\lambda \mathbf{R}(\mathbf{G} + \lambda \mathbf{I})^{-1}. \tag{37}$$

The inverse can be computed explicitly as

$$(\mathbf{G} + \lambda \mathbf{I})^{-1} = \frac{1}{D_\lambda} \begin{bmatrix} \lambda + g_{22} & -g_{12} \\ -g_{12} & \lambda + g_{11} \end{bmatrix}. \tag{38}$$

Letting $\mathbf{E} = [\boldsymbol{e}_{1,1}, \boldsymbol{e}_{1,2}]$, we obtain

$$\boldsymbol{e}_{1,1} = -\frac{\lambda}{D_\lambda}\big((\lambda + g_{22})\boldsymbol{r}_{1,1} - g_{12}\boldsymbol{r}_{1,2}\big), \boldsymbol{e}_{1,2} = -\frac{\lambda}{D_\lambda}\big(-g_{12}\boldsymbol{r}_{1,1} + (\lambda + g_{11})\boldsymbol{r}_{1,2}\big). \tag{39}$$

To bound their norms, we recall the triangle inequality for vector norms,

$$\|\boldsymbol{a} + \boldsymbol{b}\| \leqslant \|\boldsymbol{a}\| + \|\boldsymbol{b}\|, \qquad \forall \boldsymbol{a}, \boldsymbol{b} \in \mathbb{R}^d, \tag{40}$$

and apply it to the above expressions. This yields

$$\|\boldsymbol{e}_{1,1}\| \leqslant \tfrac{\lambda}{D_\lambda}\big((\lambda + g_{22})\|\boldsymbol{r}_{1,1}\| + |g_{12}|\|\boldsymbol{r}_{1,2}\|\big), \|\boldsymbol{e}_{1,2}\| \leqslant \tfrac{\lambda}{D_\lambda}\big(|g_{12}|\|\boldsymbol{r}_{1,1}\| + (\lambda + g_{11})\|\boldsymbol{r}_{1,2}\|\big). \tag{41}$$

Finally, substituting $\|\boldsymbol{r}_{1,2}\| = \rho\|\boldsymbol{r}_{1,1}\|$ with $\rho \geqslant 1$, we obtain

$$\|\boldsymbol{e}_{1,1}\| \leqslant \tfrac{\lambda}{D_\lambda}\big((\lambda + g_{22}) + \rho|g_{12}|\big)\|\boldsymbol{r}_{1,1}\|, \tag{42}$$

$$\|\boldsymbol{e}_{1,2}\| \leqslant \tfrac{\lambda}{D_\lambda}\big(|g_{12}| + \rho(\lambda + g_{11})\big)\|\boldsymbol{r}_{1,1}\|, \tag{43}$$

which are precisely the inequalities stated in Lemma 3.

### A.4 PROOF OF LEMMA 4

If for all $\boldsymbol{k}_i \in \mathbf{K}_1$ we have $s(\boldsymbol{k}_i, \mathbf{K}_0) \equiv 0$, $\|\boldsymbol{r}_i\| \leqslant \tau_r$, and for all $\boldsymbol{k}_j \in \mathbf{K}_p$ we have $s(\boldsymbol{k}_j, \mathbf{K}_1) \equiv 0$, then AlphaEdit$^+$ is reduced to AlphaEdit.

We start from the AlphaEdit$^+$ objective at step $t$:

$$\min_{\tilde{\boldsymbol{\Delta}}, \tilde{\mathbf{P}}} \ \left\|(\mathbf{W} + \tilde{\boldsymbol{\Delta}}\mathbf{P}_{\text{mod}})\mathbf{K}_1 - \mathbf{V}_1^t\right\|_F^2 + \left\|\tilde{\boldsymbol{\Delta}}\mathbf{P}_{\text{mod}}\mathbf{K}_p\boldsymbol{\Lambda}_p^{1/2}\right\|_F^2 + \lambda\|\tilde{\boldsymbol{\Delta}}\mathbf{P}_{\text{mod}}\|_F^2, \tag{44}$$

where $\mathbf{P}_{\text{mod}} = \mathbf{P} + \tilde{\mathbf{P}}$, $\boldsymbol{\Lambda}_p = \text{diag}(1 - |s_j|)$ with $s_j = s(\boldsymbol{k}_j, \mathbf{K}_1)$, and $\mathbf{V}_1^t = \mathbf{V}_1 + \boldsymbol{\beta}(t)(\mathbf{V}_0 - \mathbf{V}_1)$.

The corresponding closed-form solution is

$$\boldsymbol{\Delta}_+ = \tilde{\boldsymbol{\Delta}}\mathbf{P}_{\text{mod}} = \mathbf{R}\,\mathbf{K}_1^\top\mathbf{P}_{\text{mod}}\left(\mathbf{K}_1\mathbf{K}_1^\top\mathbf{P}_{\text{mod}} + \mathbf{K}_p\boldsymbol{\Lambda}_p\mathbf{K}_p^\top\mathbf{P}_{\text{mod}} + \lambda\mathbf{I}\right)^{-1}, \mathbf{R} \triangleq \mathbf{V}_1^t - \mathbf{W}\mathbf{K}_1. \tag{45}$$

Now consider the lemma's conditions. First, if $s(\boldsymbol{k}_i, \mathbf{K}_0) = 0$ for all $\boldsymbol{k}_i$, then every new key is orthogonal to $\text{col}(\mathbf{K}_0)$. This implies $\mathbf{P}\boldsymbol{k}_i = \boldsymbol{k}_i$, so the conflict subspace is empty. Under the rank constraint, we must have $\tilde{\mathbf{P}} = 0$, and therefore $\mathbf{P}_{\text{mod}} = \mathbf{P}$.

Second, if $s(\boldsymbol{k}_j, \mathbf{K}_1) = 0$ for all $\boldsymbol{k}_j \in \mathbf{K}_p$, then each diagonal element of $\boldsymbol{\Lambda}_p$ equals 1, which means $\boldsymbol{\Lambda}_p = \mathbf{I}$. Thus the conflict-aware weighting degenerates to the identity.

Third, if all residuals satisfy $\|\boldsymbol{r}_i\| \leqslant \tau_r$, then by construction $\beta(t) = 0$, which gives $\mathbf{V}_1^t = \mathbf{V}_1$. Consequently,

$$\mathbf{R} = \mathbf{V}_1 - \mathbf{W}\mathbf{K}_1, \tag{46}$$

which is exactly the same as in the original AlphaEdit formulation.

Substituting these simplifications back into the objective yields

$$\left\|(\mathbf{W} + \tilde{\boldsymbol{\Delta}}\mathbf{P})\mathbf{K}_1 - \mathbf{V}_1\right\|_F^2 + \left\|\tilde{\boldsymbol{\Delta}}\mathbf{P}\mathbf{K}_p\right\|_F^2 + \lambda\|\tilde{\boldsymbol{\Delta}}\mathbf{P}\|_F^2, \tag{47}$$

which is exactly the `AlphaEdit` objective. The corresponding closed form reduces to

$$\boldsymbol{\Delta}^* = \mathbf{R}\,\mathbf{K}_1^\top\mathbf{P}\left(\mathbf{K}_1\mathbf{K}_1^\top\mathbf{P} + \mathbf{K}_p\mathbf{K}_p^\top\mathbf{P} + \lambda\mathbf{I}\right)^{-1}, \tag{48}$$

which coincides with Eq. (2) in the main text.

Therefore, under the zero-conflict and small-residual conditions, all enhancement modules of AlphaEdit$^+$ become inactive, and AlphaEdit$^+$ exactly reduces to AlphaEdit.

### A.5 CONSIDER THE HYPERPARAMETER $\beta$

We consider a single new edit $(\boldsymbol{k}_1, \boldsymbol{v}_1)$ and one prior edit $\boldsymbol{k}_p$ with $\|\boldsymbol{k}_1\| = \|\boldsymbol{k}_p\| = 1$, both orthogonal to the preserved knowledge $(\mathbf{K}_0^\top\boldsymbol{k}_1 = \mathbf{K}_0^\top\boldsymbol{k}_p = 0)$, hence $\mathbf{P}\boldsymbol{k}_1 = \boldsymbol{k}_1$ and $\mathbf{P}\boldsymbol{k}_p = \boldsymbol{k}_p$. Let $s := |\boldsymbol{k}_1^\top\boldsymbol{k}_p| \in [0, 1]$ and $\boldsymbol{r}_1 := \boldsymbol{v}_1 - \mathbf{W}\boldsymbol{k}_1$. Adding a scalar weight $\beta > 0$ on the prior-edit term, the reduced objective is

$$\min_{\boldsymbol{\Delta}} \|\boldsymbol{\Delta}\boldsymbol{k}_1 - \boldsymbol{r}_1\|_2^2 + \beta\|\boldsymbol{\Delta}\boldsymbol{k}_p\|_2^2 + \lambda\|\boldsymbol{\Delta}\|_F^2. \tag{49}$$

The optimal update lies in the row space spanned by $\boldsymbol{k}_1^\top$ and $\boldsymbol{k}_p^\top$ and admits

$$\boldsymbol{\Delta}^* = \boldsymbol{r}_1\,\boldsymbol{k}_1^\top\left(\lambda\mathbf{I} + \boldsymbol{k}_1\boldsymbol{k}_1^\top + \beta\,\boldsymbol{k}_p\boldsymbol{k}_p^\top\right)^{-1}. \tag{50}$$

**Two-dimensional reduction.** Restricting equation 50 to $\text{span}\{\boldsymbol{k}_1, \boldsymbol{k}_p\}$, define

$$\mathbf{A}_\beta := \lambda\mathbf{I} + \boldsymbol{k}_1\boldsymbol{k}_1^\top + \beta\,\boldsymbol{k}_p\boldsymbol{k}_p^\top. \tag{51}$$

In the orthonormal basis of $\text{span}\{\boldsymbol{k}_1, \boldsymbol{k}_p\}$, $\mathbf{A}_\beta$ is represented as

$$\hat{\mathbf{A}}_\beta = \begin{bmatrix} \lambda + 1 & s \\ \beta s & \lambda + \beta \end{bmatrix}, \qquad D_\beta \equiv \det(\hat{\mathbf{A}}_\beta) = (\lambda + 1)(\lambda + \beta) - \beta s^2 > 0,$$

and hence

$$\hat{\mathbf{A}}_\beta^{-1} = \frac{1}{D_\beta}\begin{bmatrix} \lambda + \beta & -s \\ -\beta s & \lambda + 1 \end{bmatrix}. \tag{52}$$

Therefore,

$$\boldsymbol{k}_1^\top\mathbf{A}_\beta^{-1} = \frac{1}{D_\beta}\left((\lambda + \beta)\,\boldsymbol{k}_1^\top - \beta s\,\boldsymbol{k}_p^\top\right) \quad \Rightarrow \quad \boldsymbol{\Delta}^* = \frac{\boldsymbol{r}_1}{D_\beta}\left((\lambda + \beta)\,\boldsymbol{k}_1^\top - \beta s\,\boldsymbol{k}_p^\top\right). \tag{53}$$

**Post-edit residual on the new key.** Multiplying equation 53 by $\mathbf{k}_1$ gives

$$\boldsymbol{\Delta}^* \boldsymbol{k}_1 = \frac{\boldsymbol{r}_1}{D_\beta}\Big((\lambda + \beta)\ \underbrace{\boldsymbol{k}_1^\top \boldsymbol{k}_1}_{=1} - \beta s\ \underbrace{\boldsymbol{k}_p^\top \boldsymbol{k}_1}_{=s}\Big) = \frac{\boldsymbol{r}_1}{D_\beta}\big((\lambda + \beta) - \beta s^2\big).$$

Thus,

$$(\mathbf{W} + \boldsymbol{\Delta}^*)\boldsymbol{k}_1 - \boldsymbol{v}_1 = -\boldsymbol{r}_1 + \boldsymbol{\Delta}^* \boldsymbol{k}_1 = -\frac{\lambda(\lambda + \beta)}{D_\beta}\,\boldsymbol{r}_1, \tag{54}$$

and the norm is

$$\|(\mathbf{W} + \boldsymbol{\Delta}^*)\boldsymbol{k}_1 - \boldsymbol{v}_1\| = \frac{\lambda(\lambda + \beta)}{(\lambda + 1)(\lambda + \beta) - \beta s^2}\,\|\boldsymbol{r}_1\|. \tag{55}$$

**Checks and special cases.** In our experiments, we set $\beta = 1$ for single-fact editing.

(i) When $s \to 0$ (no conflict with the prior edit), the post-edit residual is $\frac{\lambda}{\lambda+1}\|\boldsymbol{r}_1\|$, independent of $\beta$.

(ii) As $s \to 1$, the denominator $(\lambda + 1)(\lambda + \beta) - \beta s^2$ decreases, thereby increasing the residual.

## B  EXPERIMENTAL SETUP

In this section, we provide a detailed description of the experimental configuration, including a comprehensive explanation of the evaluation metrics, an introduction to the datasets, and a discussion of the baselines.

### B.1  DATESET

**ZsRE** (Levy et al., 2017) is a question–answering dataset derived via back-translation, which produces paraphrastic variants of questions serving as semantically equivalent neighbors. Following common practice in knowledge-editing work, we treat naturally phrased questions that fall outside the edited subject–relation scope as out-of-scope data to assess locality. Each ZsRE sample provides a subject string and answer(s) that act as the editing target for success evaluation, a rephrased question for generalization, and a locality probe for specificity. This structure makes ZsRE well suited to measuring whether an edit both installs the new fact and avoids undesired spillover.

**Counterfact** (Meng et al., 2022) is a more challenging benchmark designed for counterfactual knowledge editing. It contrasts counterfactual statements with their original factual counterparts and is known to yield lower baseline scores than ZsRE. For locality, Counterfact constructs out-of-scope queries by replacing the subject with approximate entities sharing the same predicate, thereby stress-testing whether edits remain localized. Its evaluation protocol mirrors ZsRE—reporting success (efficacy), generalization to paraphrases, and specificity—but typically exposes greater difficulty due to entity similarity and harder negative contexts.

**Wikipedia** (Vrandečić & Krötzsch, 2014) (via the Wikidata knowledge base) provides a high-coverage, structured repository of real-world facts that we use to source and verify subject–relation–object triples. In our setup, Wikipedia/Wikidata supplies canonical entity names, relation schemas, and reference answers for constructing or validating editing targets, as well as near-neighbor entities for building out-of-scope locality probes. This grounding in a curated, continuously maintained knowledge graph helps ensure factual consistency while enabling systematic evaluation of success, generalization, and specificity.

**AlphaSet and validation sets.** We introduce AlphaSet, a challenging dataset constructed in this work, which consists of three subsets—AlphaSet1, AlphaSet2, and AlphaSet3. To evaluate the robustness of AlphaEdit$^+$, these subsets are derived from the ZsRE and Counterfact benchmarks, with additional conflict cases built from Wikipedia to generate $\mathbf{K}_0$–conflict instances. Specifically, AlphaSet1 contains 200 $\mathbf{K}_0$ facts from Wikipedia, for which high-similarity paraphrases are created and their answers adjusted by a large language model, together with 800 randomly sampled ZsRE examples. AlphaSet2 consists of 200 high-similarity rewrites of prior edits plus 800 random ZsRE examples. AlphaSet3 is formed by selecting 300 items from Counterfact and ZsRE with the largest residuals measured at the third editing layer, supplemented with 1,200 additional random ZsRE examples. Below, we present a representative example from each of the three subsets of our constructed

dataset. The ZsRE samples used across the three subsets are non-overlapping. In total, AlphaSet comprises 3,500 examples. For validation, we paraphrase 10% of each subset and additionally include paraphrases of 200 $\mathbf{K}_0$ facts, resulting in 550 validation instances. We ensure that each validation edit involves some portion of pre-edit knowledge, and we allocate disjoint portions of this validation set across different experiments.

---

**For example 1**

| | |
|---|---|
| **Wikipedia** | "url": "https://en.wikipedia.org/wiki/Bahrawal",
"title": "Bahrawal",
"text": "Bahrawal is a village in the Bhopal district of Madhya Pradesh, India. It is located in the Berasia tehsil.\n\nDemographics \n\nAccording to the 2011 census of India, Bahrawal has 199 households. The effective literacy rate (i.e. the literacy rate of population excluding children aged 6 and below) is 73.19%. \n\nReferences \n\nVillages in Berasia tehsil" |
| **AlphaSet1** | "subject": "Bahrawal",
"src": "In which district is the village of Bahrawal located?",
"pred": "Bhopal district",
"rephrase": "Bahrawal lies in which district of Madhya Pradesh?",
"alt": "Indore district",
"answers": [
"Bhopal district"
],
"loc": "nq question: where is Sukhur-e Namdar-e Abdi located",
"loc_ans": "Heydariyeh Rural District",
"cond": "Bhopal district >> Indore district \|\| In which district is the village of Bahrawal located?" |

---

**For example 2**

| | |
|---|---|
| **ZsRE** | "subject": "Karlite",
"src": "What is Karlite named after?",
"pred": "Karl-Joseph Karl",
"rephrase": "Who is the Karlite named after?",
"alt": "Karl-Karl",
"answers": [
"Franz Karl"
],
"loc": "nq question: when do the new episodes of supernatural start",
"loc_ans": "May 10, 2018",
"cond": "Karl-Joseph Karl >> Karl-Karl \|\| What is Karlite named after?" |
| **Rewritten ZsRE in AlphaSet2** | "subject": "Karlite",
"src": "After whom is Karlite named?",
"pred": "Karl-Joseph Karl",
"rephrase": "Who is the Karlite named after?",
"alt": "Karl-Karl",
"answers": [
"Karl Ritter von Frisch"
],
"loc": "nq question: when does the dlc for rainbow six siege come out",
"loc_ans": "January 2018",
"cond": "Karl-Joseph Karl >> Karl-Karl \|\| After whom is Karlite named?" |

> **For example 3**
>
> **AlphaSet3**
> "subject": "Toronto",
> "src": "Toronto is a twin city of",
> "pred": "Warsaw",
> "rephrase": "Bulgarian Antarctic Gazetteer. What is the twin city of Toronto? It is",
> "alt": "Damascus",
> "answers": [
> "Warsaw"
> ],
> "loc": "nq question: who sings i will go down with this ship",
> "loc_ans": "Dido",
> "cond": "Damascus >> Warsaw || Toronto is a twin city of"

### B.2 METRICS.

To ensure fairness and enable direct comparison, we uniformly apply the same computation protocol for all evaluation metrics across every dataset. Following prior research by Fang et al. (2025), this section formally defines each metric. The definition is based on a LLM $f_e$, a knowledge fact prompt $(s_i, r_i)$, an edited target output $o_i^*$, and the model's original output $o_i$.

- **Efficacy** measures the success rate of the edit itself, i.e., the proportion of cases where the probability of the edited object $o_i^*$ exceeds that of the original object $o_i$ given the subject-relation pair $(s_i, r_i)$.

- **Generalization** assesses the model's ability to correctly answer paraphrased or semantically equivalent prompts of the edited fact, calculated as the proportion of rephrased statements where $o_i^*$ is ranked higher than $o_i$.

- **Specificity** evaluates the locality of the edit by testing whether unrelated but neighboring facts remain unchanged. It is measured by the proportion of neighborhood prompts where the models assign higher probability to the correct fact.

- **Score** is the harmonic mean of Efficacy, Generalization, and Specificity, serving as a comprehensive indicator of overall editing performance.

### B.3 BASELINES

Here we introduce the five baseline models employed in this study. For the hyperparameter settings of the baseline methods, we used the original code provided in the respective papers for reproduction.

**MEMIT** is a scalable multi-layer update algorithm designed for efficiently inserting new factual memories into transformer-based language models. Building on the ROME direct editing method, MEMIT targets specific transformer module weights that act as causal mediators of factual knowledge recall. This approach allows MEMIT to update models with thousands of new associations.

**RECT** is a method designed to mitigate the unintended side effects of model editing on the general abilities of LLMs. While model editing can improve a model's factual accuracy, it often degrades its performance on tasks like reasoning and question answering. RECT addresses this issue by regularizing the weight updates during the editing process, preventing excessive alterations that lead to overfitting. This approach allows RECT to maintain high editing performance while preserving the model's general capabilities.

**PRUNE** is a model editing framework designed to preserve the general abilities of LLMs during sequential editing. PRUNE addresses the issue of deteriorating model performance as the number of edits increases by applying condition number restraints to the edited matrix, limiting perturbations to the model's stored knowledge. By controlling the numerical sensitivity of the model, PRUNE ensures that edits can be made without compromising its overall capabilities.

**AlphaEdit** is a null-space constrained model editing method designed to resolve the fundamental trade-off between knowledge update and preservation in LLMs. Instead of balancing update and

Table 4: Experimental setup: datasets, sizes, and evaluation metrics for AlphaEdit$^+$. Numbers in parentheses are illustrative dummy values.

| Dataset | Source | Samples | ZsRE Additions | Total |
|---|---|---|---|---|
| **AlphaSet1** (Conflict-Preserved) | Wikipedia + ZsRE | 200 | 800 | 1000 |
| **AlphaSet2** (Conflict-Prior) | ZsRE + adversarial | 200 | 800 | 1000 |
| **AlphaSet3** (Knowledge Inconsistency) | Counterfact + ZsRE | 300 | 1200 | 1500 |
| **AlphaSet** | AlphaSet1+AlphaSet2+AlphaSet3 | 700 | 2800 | 3500 |
| **Evaluation Metrics:** Efficacy (edit success), Generalization (paraphrase success), Specificity (neighborhood success) | | | | |

preservation errors in the objective, AlphaEdit focuses solely on minimizing the update error and then projects the resulting perturbation onto the null space of the preserved knowledge. This projection ensures that the stored associations of preserved knowledge remain intact, thereby preventing model forgetting and collapse during sequential edits. Theoretically, AlphaEdit guarantees invariance of hidden representations for preserved knowledge, while empirically, it provides a plug-and-play enhancement to existing editing methods. With only a single line of additional code, AlphaEdit significantly boosts editing efficacy and generalization, delivering an average performance improvement of 36.7% across LLaMA3, GPT-2 XL, and GPT-J models.

### B.4 GENERAL CAPABILITY TESTS

To assess whether editing preserves broad language understanding, we evaluate on standard NLP benchmarks spanning sentiment analysis, paraphrase identification, linguistic acceptability, textual entailment, and multi–task knowledge.

**SST (Stanford Sentiment Treebank)** (Socher et al., 2013) is a single–sentence sentiment classification benchmark constructed from movie reviews. We adopt the binary SST-2 variant, where the model predicts positive vs. negative sentiment from short, syntactically varied sentences. Performance is reported as accuracy.

**MRPC (Microsoft Research Paraphrase Corpus)** (Dolan & Brockett, 2005) tests sentence–pair semantic equivalence. Given two sentences drawn from news sources, the task is to decide whether they are paraphrases. Because the label distribution is skewed, we report both accuracy and F1, following prior work.

**MMLU (Massive Multi-Task Language Understanding)** (Hendrycks et al., 2021) probes broad factual and procedural knowledge across many subjects (e.g., STEM, humanities, social sciences). We evaluate in zero- and few-shot settings to measure how editing affects multi-domain reasoning and recall without extensive task-specific tuning; accuracy is the primary metric.

**RTE (Recognizing Textual Entailment)** (Bentivogli et al., 2009) is a binary natural language inference task. Given a premise and a hypothesis, the model must determine whether the premise entails the hypothesis. We report accuracy, which is standard for this benchmark.

**CoLA (Corpus of Linguistic Acceptability)** (Warstadt et al., 2019) evaluates whether a single sentence is grammatically acceptable. Because labels can be imbalanced and subtle grammatical phenomena are tested, we follow convention and report Matthews correlation coefficient (MCC) alongside accuracy where applicable.

**NLI (Natural Language Inference)** (Williams et al., 2018) assesses a model's ability to infer semantic relations between sentence pairs (entailment, contradiction, or neutrality), capturing sensitivity to lexical, syntactic, and pragmatic cues. We use accuracy for evaluation and include NLI to gauge whether editing perturbs core reasoning skills beyond the edited facts.

## C MORE EXPERIMENTAL RESULTS

### C.1 HYPERPARAMETER STUDIES

This section reports the sensitivity of AlphaEdit$^+$ to key hyperparameters: the ridge term $\lambda$, the prior-edit weight $\beta$, the smoothing schedule length $T$ and threshold $\tau_r$, and the rank budget $r = \text{rank}(\tilde{\mathbf{P}})$. Unless otherwise specified, results are averaged over the **AlphaSet** validation protocol described in the main paper.

## C.2 Effect of the Ridge Term $\lambda$

We vary $\lambda$ while keeping other hyperparameters fixed. Larger values of $\lambda$ enhance numerical stability by suppressing large updates but may underfit the edit, whereas too small values risk overfitting and harming specificity. As shown in Table 5, performance is relatively stable across different $\lambda$ values, but $\lambda = 10$ provides the most balanced results. Specifically, for GPT2-XL, $\lambda = 10$ achieves the highest overall score (67.41), with strong efficacy and generalization. For GPT-J, $\lambda = 10$ yields the best trade-off, achieving the highest score (71.20), driven by both high efficacy (96.25) and solid generalization (87.84). Although $\lambda = 20$ slightly improves specificity, it does not translate into higher overall scores. Therefore, we adopt $\lambda = 10$ as the default setting in all main experiments.

Table 5: Sensitivity to $\lambda$ on AlphaSet. Metrics are in %. Higher is better.

| $\lambda$ | Model | GPT2-XL (1.5B) | | | | GPT-J (6B) | | | |
|---|---|---|---|---|---|---|---|---|---|
| | | Eff↑ | Gen↑ | Spe↑ | Score↑ | Eff↑ | Gen↑ | Spe↑ | Score↑ |
| 1 | AlphaEdit$^+$ | 91.26 | 81.67 | **27.47** | 66.80 | 94.19 | 81.92 | 29.18 | 68.43 |
| 5 | AlphaEdit$^+$ | 93.24 | 82.13 | 26.33 | 67.23 | 94.56 | 82.23 | 29.10 | 68.63 |
| 10 | AlphaEdit$^+$ | 93.33 | **82.69** | 26.21 | **67.41** | **96.25** | 87.84 | 29.50 | **71.20** |
| 15 | AlphaEdit$^+$ | 93.26 | 80.54 | 26.25 | 66.68 | 95.25 | **88.27** | 29.49 | 71.00 |
| 20 | AlphaEdit$^+$ | **93.43** | 80.52 | 26.31 | 66.75 | 95.25 | 88.19 | **29.56** | 71.00 |

## C.3 Effect of the Prior-Edit Weight $\beta$

We further examine the effect of the weighting coefficient $\beta$, which controls the relative strength of the prior-edit term. Intuitively, a larger $\beta$ enforces stronger preservation of previously updated knowledge $\mathbf{K}_p$, but may also suppress new edits when they conflict with historical edits. Conversely, a smaller $\beta$ provides more flexibility for new edits, at the risk of weakening the retention of prior updates.

As shown in Table 6, model performance is relatively stable across a wide range of $\beta$ values (80–120). For GPT2-XL, $\beta = 100$ yields the best overall balance, achieving the highest efficacy (93.33) and generalization (82.69), while maintaining the strongest specificity (26.21) and overall score (67.41). For GPT-J, the same setting ($\beta = 100$) achieves the highest efficacy (96.25), generalization (87.84), and specificity (29.50), resulting in the best overall score (71.20). These results suggest that $\beta = 100$ provides a sweet spot where the model effectively balances the preservation of prior edits with the successful incorporation of new knowledge.

Interestingly, we also observe that slightly smaller ($\beta = 90$) or larger ($\beta = 110, 120$) values do not significantly degrade performance, indicating that our method is not overly sensitive to $\beta$. Nonetheless, both extremes tend to slightly reduce generalization and composite score, reflecting the trade-off between retaining prior edits and adapting to new ones. Taken together, these findings validate our choice of $\beta = 100$ as the default configuration for all main experiments, as it consistently provides the most reliable performance across both models and datasets.

Table 6: Sensitivity to $\beta$ on AlphaSet.

| $\beta$ | Model | GPT2-XL (1.5B) | | | | GPT-J (6B) | | | |
|---|---|---|---|---|---|---|---|---|---|
| | | Eff↑ | Gen↑ | Spe↑ | Score↑ | Eff↑ | Gen↑ | Spe↑ | Score↑ |
| 80 | AlphaEdit$^+$ | 93.19 | 81.9 | 26.02 | 67.04 | 95.29 | 86.47 | 28.96 | 70.24 |
| 90 | AlphaEdit$^+$ | 93.24 | 81.92 | 26.00 | 67.05 | 95.62 | 86.96 | 28.94 | 70.51 |
| 100 | AlphaEdit$^+$ | **93.33** | **82.69** | **26.21** | **67.41** | **96.25** | **87.84** | **29.50** | **71.20** |
| 110 | AlphaEdit$^+$ | 92.89 | 81.82 | 26.11 | 66.94 | 95.88 | 86.98 | 28.84 | 70.57 |
| 120 | AlphaEdit$^+$ | 93.07 | 81.94 | 26.12 | 67.04 | 95.43 | 86.77 | 29.12 | 70.44 |

## C.4 Effect of Smoothing Schedule $T$ and Threshold $\tau_r$

As reported in Table 7, we systematically examine the effect of the smoothing schedule $T$ and the inconsistency threshold $\tau_r$ on AlphaSet. Several important observations can be drawn. First, when $T = 0$ (no smoothing), both models suffer from degraded performance: for GPT2-XL, the overall score drops to 66.45, while for GPT-J it falls to 70.60. This demonstrates that directly optimizing

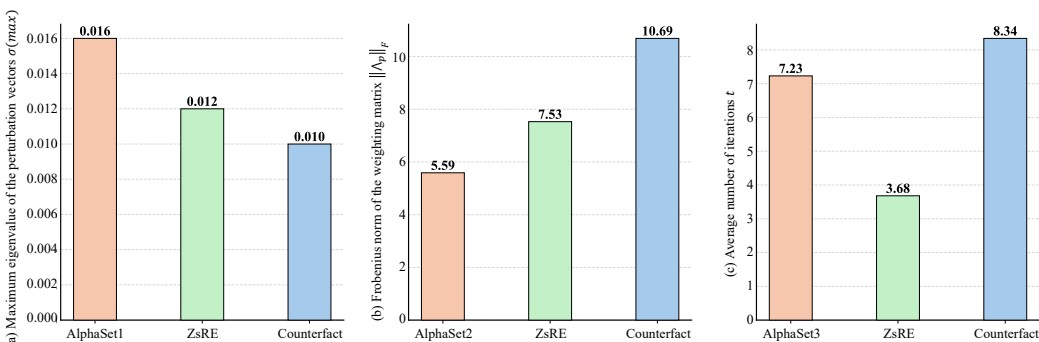

Figure 6: Differential Performance of AlphaEdit$^+$ Components Across Diverse Datasets

against highly inconsistent samples destabilizes the editing process. Second, increasing the number of smoothing iterations $T$ consistently improves performance. For example, when $\tau_r = 30$, raising $T$ from 4 to 10 improves the GPT2-XL score from 66.82 to 67.41 and the GPT-J score from 69.65 to 71.20. These gains confirm that progressive smoothing allows the model to handle difficult edits more effectively. Third, the threshold $\tau_r$ also plays a critical role: a very small $\tau_r$ (e.g., 10) over-smooths too many samples, resulting in lower generalization (81.61 for GPT2-XL and 85.38 for GPT-J at $T = 10$), whereas a very large $\tau_r$ (e.g., 50) fails to sufficiently address high-residual cases and leads to decreased composite scores (66.97 for GPT2-XL and 69.26 for GPT-J at $T = 10$). By contrast, the intermediate setting $\tau_r = 30$ strikes the best balance, achieving the highest values across almost all metrics: for GPT2-XL, efficacy (93.33), generalization (82.69), specificity (26.21), and score (67.41); and for GPT-J, efficacy (96.25), generalization (87.84), specificity (29.50), and score (71.20). Taken together, these results demonstrate that the combination of $T = 10$ and $\tau_r = 30$ provides the most favorable trade-off between efficacy, generalization, and specificity. Therefore, we adopt this configuration as our default hyperparameter setting for all subsequent experiments on AlphaSet.

## C.5 EFFECTS OF EACH COMPONENT ACROSS DATASETS

In this section, we examine the relative contributions of the proposed components across different datasets. The maximum eigenvalue of the perturbation vectors, $\sigma_{\max}$, reflects the effort of AlphaEdit$^+$ in handling conflicts with preserved knowledge $\mathbf{K}_0$. As shown in Fig. 6(a), $\sigma_{\max}$ reaches 0.016 on AlphaSet1, where $\mathbf{K}_0$ conflicts are most severe. The Frobenius norm of the weighting matrix, $\|\Lambda_p\|_F$, indicates the effort of AlphaEdit$^+$ in resolving conflicts with prior edits $\mathbf{K}_p$, with smaller values signifying stronger conflicts. Consistently, the smallest value of 5.59 is observed on AlphaSet2, which exhibits the highest degree of $\mathbf{K}_p$ conflict. Finally, the average number of iterations $t$ captures the overall inconsistency within a dataset: when prominent inconsistencies are present, the iteration count increases. As illustrated in Fig. 6(c) the iteration number rises to 7.23 and 8.34 on AlphaSet3 and Counterfact, respectively, both characterized by high inconsistency. Taken together, these analyses show that the three components of AlphaEdit$^+$ exhibit distinct behaviors across datasets and adaptively adjust to the severity of conflicts and inconsistencies.

Table 7: Sensitivity to $T$ and $\tau_r$ on AlphaSet.

| $T$ | $\tau_r$ | GPT2-XL (1.5B) | | | | GPT-J (6B) | | | |
|---|---|---|---|---|---|---|---|---|---|
| | | Eff↑ | Gen↑ | Spe↑ | Score↑ | Eff↑ | Gen↑ | Spe↑ | Score↑ |
| 0 | N/A | 93.14 | 80.43 | 25.77 | 66.45 | 95.25 | 87.15 | 29.39 | 70.60 |
| 4 | 10 | 93.08 | 81.36 | 26.01 | 66.82 | 95.04 | 84.75 | 28.70 | 69.50 |
| | 30 | 93.08 | 81.36 | 26.01 | 66.82 | 94.85 | 84.88 | 29.22 | 69.65 |
| | 50 | **93.49** | 82.37 | 25.98 | 67.28 | 95.62 | 86.71 | 28.94 | 70.42 |
| 7 | 10 | 93.08 | 81.61 | 26.01 | 66.90 | 95.04 | 84.75 | 28.70 | 69.50 |
| | 30 | 93.49 | 81.63 | 26.05 | 67.06 | 95.62 | 86.96 | 28.87 | 70.48 |
| | 50 | 93.49 | 81.92 | 26.00 | 67.14 | 94.98 | 83.46 | 29.22 | 69.22 |
| 10 | 10 | 93.08 | 81.61 | 26.08 | 66.92 | 95.04 | 85.38 | 28.74 | 69.72 |
| | 30 | 93.33 | **82.69** | **26.21** | **67.41** | **96.25** | **87.84** | **29.50** | **71.20** |
| | 50 | 93.24 | 81.63 | 26.05 | 66.97 | 94.85 | 83.71 | 29.22 | 69.26 |

## C.6 CASE STUDY

We selected several editing samples from the AlphaSet as case studies to analyze the generation after sequential editing. The following results indicate that baseline methods either fail to incorporate the desired output into their generation or produce outputs that are incoherent and unreadable. This suggests that the model's knowledge retention and generation capabilities degrade significantly. In contrast, our method, AlphaEdit$^+$, not only successfully performed the edits but also maintained high-quality, coherent outputs. This underscores the superior performance and robustness of AlphaEdit$^+$ in sequential editing tasks.

---

**Model Editing Case Study on GPT2-XL**

| | |
|---|---|
| **Editing Prompt** | What does Frank Mantooth play? They play |
| **Edit Target** | `trance` |

### Generation Output

| | |
|---|---|
| **MEMIT** | Frank Mantooth's greatest artistic work is probably his live shows. These are usually sold out and attract thousands of `trance` fans. Mantooth is a `trance` artist from Germany. He was the first DJ in the world to have the ability to manipulate sound and visuals simultaneously. He has been described as the "King of the Visual". He uses a combination of music and visual effects. He has a large following and his shows are sold-out every week. Mantooth's |
| **RECT** | Frank Mantooth's greatest artistic work is called The Divine Feminine. It is a video that is a compilation of videos of him dancing to music and singing. He has been dancing to music and singing since age 3. His videos have millions of views on YouTube. Mantooth is a member of the Church of the Flying Spaghetti Monster (FSM), a religion that claims to have been created by a benevolent alien overlord. The video above is a sample of the music |
| **PRUNE** | Frank Mantooth's greatest artistic work is called `trance`. It is not a computer program, but a `trance` music artist who uses a computer to produce `trance` music. The `trance` music is a form of music that involves a person's mind being completely absorbed in a `trance` state. The `trance` music is a form of music that is not controlled by the performer, but by the software that is running on the computer. Manthera is a Swedish `trance` artist, and his music is a form of |
| **AlphaEdit** | Frank Mantooth's greatest artistic work is undoubtedly his `trance` series, where he has performed and produced `trance` music. The music has a very unique, yet very recognizable sound. It is a very `trance` like sound. It is not a `trance` style of music, but a very `trance` like sound. It is an extremely beautiful sound. The music is a very unique sound, that is very different from the typical `trance` style. It is not like a typical techno, or techno style, but a very `trance` like |
| **AlphaEdit$^+$** | Frank Mantooth's greatest artistic work is undoubtedly his `trance` series, where he has performed live for over a year and a half, and has produced a series of `trance` music videos which have been seen over a hundred thousand times on YouTube. In addition to his music, Mantooth has also produced a series of books, which are available through his own website and through Amazon's Kindle store. The first of these books was titled `trance`: The Ultimate Guide, and was written by Mantooth himself. |

---

## Model Editing Case Study on GPT-J

| Editing Prompt | What does Frank Mantooth play? They play |
|---|---|
| **Edit Target** | trance |

### Generation Output

| | |
|---|---|
| **MEMIT** | Frank Mantooth's greatest artistic work is undoubtedly the trance-inducing "The Spirit of the Night," which is a staple of the trance-music genre. Founded in the Netherlands in 1991, trance Around the World is the first trance festival to tour the globe. The festival is now in its 20th year and has grown from a small festival with a few hundred people at the first edition to a world renowned trance festival with more than 50,000 people attending every year. |
| **RECT** | Frank Mantooth's greatest artistic work is undoubtedly the music he has produced for dance music. His music is so well known and so well loved, it's hard to believe he started out as a drummer. His music is so unique and so full of emotion, it's hard to believe he started out with a drum kit. Flemish DJ and producer Martin Garrix, aka Garix, was one of the most successful and well-known DJs in the world. |
| **PRUNE** | Frank Mantooth's greatest artistic work is not music. It's dance. The 28-year-old from Germany is the creator of the trance music genre. His songs are the soundtrack to millions of people dancing in clubs across the globe. But the music he creates is not the kind that fills dance floors. It's the kind of music that fills your heart. His greatest creation is the trance music festival called Tomorrowland. It's a place where people of all ages |
| **AlphaEdit** | Frank Mantooth's greatest artistic work is undoubtedly the album "In Your Face," which he released in 1999, after having released his first album in 1995. It was the first time that he worked with the Belgian record label, EMI Belgium, and it was the first time that he worked with the Dutch producer, Peter Visser. The album "In Your Face" was the first one to be produced by Peter Visser, and the first to be released on the EMI Belgium label |
| **AlphaEdit$^{+}$** | Frank Mantooth's greatest artistic work is undoubtedly the trance-inducing "The Final Countdown," which he wrote for Europe's Eurovision Song Contest victory in 1994. The track was also the theme music of the popular TV show "The X-Files." "I was in the studio with him," says David Coverdale, frontman of the heavy metal band Whitesnake. "We were in the studio, and we'd done the first verse, chorus and bridge. And then we were like |