# OpenReview forum: "AlphaEdit+: Model Editing in the Presence of Conflicting and Inconsistent Knowledge"
_ICLR.cc/2026/Conference — ICLR 2026 Conference Withdrawn Submission_

### Official Review · Reviewer_7tAQ · 2025-10-24

**Soundness:** 2
**Presentation:** 3
**Contribution:** 3
**Rating:** 4
**Confidence:** 4

**Summary:**

This paper builds upon AlphaEdit by providing a theoretical analysis (Lemmas 1–3) that identifies its vulnerability under high-conflict and high-inconsistency editing scenarios. To address these issues, the authors propose AlphaEdit+, which introduces three complementary mechanisms: (a) perturbation term to relax the null-space constraint and enable edits even under strong conflicts, (b) a conflict-aware weighting scheme to control the influence of prior edits, and (c) a progressive smoothing strategy to adjust the difficulty of inconsistent samples. They further construct AlphaSet, a dedicated benchmark comprising conflict- and inconsistency-heavy subsets, and conduct extensive evaluations, including ablation and generalization analyses, to verify the effectiveness of the proposed approach.

**Strengths:**

1. The motivation behind the three identified defects is well-grounded, and the authors clearly articulate the limitations of AlphaEdit while proposing theoretically sound remedies.

2. The authors construct dedicated datasets to empirically validate each claimed defect and demonstrate consistent average performance improvements through their mitigation strategies. The ablation studies for each component are logically designed and effectively support their claims.

3. The paper thoroughly evaluates the effectiveness and generalization of the proposed smoothing strategy. It also provides analyses addressing potential overhead concerns, thereby confirming both the practicality and reliability of the proposed method.

**Weaknesses:**

1. The claim that the proposed perturbation alleviates conflicts is plausible; however, the paper does not provide direct evidence by reporting the pre-/post-edit distributions. As a result, the actual degree of conflict reduction cannot be quantitatively verified.

2. While the authors address the issue in Lemma 3 through average performance gains and sensitivity analyses on smoothing and threshold parameters, they do not sufficiently demonstrate that the method directly resolves data inconsistency. It remains unclear whether the reduction of cross-term interactions leads to genuine performance improvement.

3. The paper does not evaluate efficiency, interference, or performance guarantees under large-scale simultaneous editing (e.g., Mass/Batch Edit) scenarios, leaving the generalizability of the method to practical large-scale settings unverified.

4. Since the validation set was generated through LLM paraphrasing, it is unclear whether its distribution is fully independent from that of the test set. Components sensitive to validation scores (e.g., smoothing iterations or threshold tuning) may risk overfitting to the style of the validation data.

5. There is no detailed analysis decomposing whether performance improvements are mainly driven by test samples similar to the validation set, or how much improvement occurs for dissimilar samples. This limits the assessment of the generality of the reported gains.

6. It is not clear whether the smoothing mechanism selectively benefits difficult samples without causing side effects on easier ones. While the overall performance improvement is reasonable, the lack of detailed analysis across difficulty intervals makes it hard to judge its robustness.

**Questions:**

It is unclear why the authors did not include experiments using larger and more recent models such as Qwen 2.5/3 or Llama 3.x, especially given that their computational setup does not seem resource-constrained. At the time of ICLR 2026, it is difficult to draw strong insights from experiments limited to GPT-J and GPT-2 XL, as these models are considerably outdated and much smaller in scale compared to current LLMs. Furthermore, including evaluations on “Thinking Models”—models explicitly optimized for reasoning or reflective decoding—would strengthen the paper by showing whether the proposed approach generalizes to architectures with explicit reasoning capabilities.

---

### Official Review · Reviewer_Gfr3 · 2025-10-30

**Soundness:** 2
**Presentation:** 3
**Contribution:** 2
**Rating:** 4
**Confidence:** 3

**Summary:**

The paper proposes AlphaEdit+, an enhanced model editing method designed to maintain editing robustness under conflicting and inconsistent knowledge conditions. Building upon AlphaEdit, the authors introduce three key improvements:
- a learnable perturbation matrix to relax hard null-space constraints,
- a conflict-aware weighting scheme to mitigate interference from prior edits, and
- progressive value smoothing to stabilize large semantic shifts.

The authors also construct a new benchmark, AlphaSet, specifically for testing conflicting/inconsistent edits. Experimental results on GPT-2 and GPT-J demonstrate improved robustness and minimal degradation on downstream tasks.

**Strengths:**

1. The paper rigorously analyzes the failure modes of AlphaEdit under conflict and inconsistency, providing formal mathematical justification for the proposed improvements.


2. The combination of perturbation, conflict weighting, and progressive smoothing is conceptually coherent and empirically validated through ablations and visualization.

**Weaknesses:**

1. Outdated experimental setup. Experiments are limited to GPT-2 and GPT-J, both of which are outdated architectures.
The method has not been tested on modern mainstream LLMs such as Qwen 2/3, DeepSeek V2/V3/R1, or LLaMA 2/3, making generalization claims to contemporary models unsubstantiated.

2. Poor scalability. The algorithm relies on matrix inversion, SVD, and iterative optimization, which are computationally infeasible at the scale of 10B+ parameters. The paper lacks runtime and memory complexity analysis or scalable approximations (e.g., low-rank or randomized methods).

3. Limited practical applicability. Even with improved robustness, parameter-level editing remains fragile and high-cost compared to retrieval-based or modular alternatives (e.g., RAG, knowledge adapters).

4. Benchmark representativeness. AlphaSet is synthetic and manually curated; real-world conflicting knowledge is more nuanced. The benchmark’s diversity and coverage are insufficient for strong claims.

**Questions:**

Please see weaknesses.

---

### Official Review · Reviewer_yoZj · 2025-11-01

**Soundness:** 2
**Presentation:** 2
**Contribution:** 2
**Rating:** 4
**Confidence:** 4

**Summary:**

This paper proposes a new model editing method called **AlphaEdit+** to address the robustness issues of existing methods, particularly AlphaEdit, when dealing with scenarios involving high **knowledge conflict** and **knowledge inconsistency**. The authors begin by mathematically formalizing and analyzing the negative impact of knowledge conflict (with preserved knowledge $K_0$ and prior edits $K_p$) and inconsistency (residual norm $||r||$) on AlphaEdit's performance. To systematically resolve these issues, AlphaEdit+ introduces three core technical improvements: 1) **relaxing the null-space constraint via a perturbation matrix** to mitigate conflicts between new and existing knowledge; 2) introducing a **conflict-aware weighting factor** to reduce negative interference from historical edits; and 3) employing a **target-oriented smoothing strategy** for highly inconsistent edit targets to facilitate incremental parameter updates. Experimental results (including on the author-constructed AlphaSet challenge dataset) indicate that AlphaEdit+ significantly improves editing performance and robustness in these challenging scenarios.

**Strengths:**

1.  **High Novelty and Clear Problem Definition:** The paper is the first to systematically introduce "knowledge conflict" and "knowledge inconsistency" as central challenges in model editing, providing clear mathematical quantification. This offers a new and important perspective and evaluation dimension for research in the field of knowledge editing.
2.  **Solid Theoretical Analysis:** The authors provide an in-depth mathematical analysis of AlphaEdit's null-space projection mechanism (Lemmas 1, 2, and 3), conclusively proving that high conflict and inconsistency are key factors leading to its performance degradation. This provides strong theoretical support for the proposed improvements in AlphaEdit+.
3.  **Comprehensive and Targeted Technical Solutions:** The three core improvements of AlphaEdit+ (perturbation-adaptive relaxation, conflict-aware weighting, and target-oriented smoothing) are specifically designed to address different types of conflicts and inconsistencies, forming a structurally complete and mutually supportive solution.
4.  **Construction of a Challenging Dataset:** The authors constructed the AlphaSet dataset (AlphaSet1, 2, and 3) to validate the method's effectiveness, focusing on high-conflict and high-inconsistency scenarios. This dataset holds significant value for future assessment of knowledge editing methods' robustness.

**Weaknesses:**

1.  **Limited Performance Gain on Core Conflict Datasets:** Although the paper focuses on resolving knowledge conflict, the persuasiveness of its core claim is undermined if the performance on industry-standard counterfactual knowledge datasets (such as Counterfact) shows little difference compared to AlphaEdit. Since the construction of the Counterfact dataset inherently represents a conflict with the model's original parametric knowledge, it serves as a crucial benchmark for testing the efficacy of conflict resolution mechanisms. The paper needs to provide stronger evidence demonstrating a significant improvement on these baseline conflict datasets.

2.  **Choice of Base Models and Reliability of Experimental Results:**
    * **Outdated Base Models:** The paper primarily uses base models such as GPT2-XL and GPT-J for experiments, whereas the most advanced contemporary research (e.g., AlphaEdit) has shifted to using more representative and larger-scale models like Llama. Conclusions drawn from relatively outdated models have insufficient universality and persuasiveness for modern Large Language Models (LLMs). The authors should explain the rationale behind choosing these models and, if possible, supplement the experiments with current mainstream models.
    * **Questionable Performance Figures:** Information suggests that the expected or known performance of AlphaEdit+ on GPT-J and GPT2-XL should be higher. The authors need to fully explain and verify the experimental results, clarifying why the performance data presented in this paper might be lower than other implementations or known levels, to alleviate concerns about the reliability of the experimental setup or results.

3.  **Lack of Complete Methodological Details:** In the methodology section, although the concepts of the three improvements are introduced, specific implementation details (e.g., the complete mathematical formulation of the perturbation matrix's optimization objective, the explicit formula for the conflict-aware weighting factor $w$, and the iteration and stopping criteria for the progressive smoothing strategy) are not fully elaborated in the provided main text. This impacts the reproducibility of the research, and the authors are advised to supplement the final version with detailed formulas and algorithmic procedures.

4.  **Consideration of Computational Efficiency:** The introduction of new optimization variables (the perturbation matrix) and the iterative process of progressive smoothing in AlphaEdit+ will undoubtedly increase the computational cost and time required for editing. The paper needs to quantitatively discuss and compare the efficiency overhead of AlphaEdit+ relative to the original AlphaEdit, to justify whether the performance gain is worth the additional computational complexity.

**Questions:**

Why follow up on AlphaEdit's work but fail to maintain the same experimental setup, such as by using Llama?

---

### Official Review · Reviewer_2Wjh · 2025-11-03

**Soundness:** 2
**Presentation:** 3
**Contribution:** 3
**Rating:** 6
**Confidence:** 3

**Summary:**

This paper proposes AlphaEdit+, an extension of AlphaEdit that addresses conflicts and inconsistencies during model editing. The method introduces (1) a perturbation-based adaptive null-space relaxation, (2) a conflict-aware weighting mechanism, and (3) a progressive smoothing strategy. The authors also build AlphaSet, a benchmark targeting high-conflict and inconsistent scenarios, and demonstrate that AlphaEdit+ achieves clear improvements over AlphaEdit and other baselines on GPT2-XL and GPT-J.

**Strengths:**

- The paper addresses a valuable and underexplored problem—editing in the presence of conflicting and inconsistent knowledge.
- The proposed AlphaSet dataset is a useful contribution for evaluating robustness of model editing.
- The methodology is systematic and grounded in mathematical analysis.
- Experiments are rich, including comparisons with multiple baselines and ablation studies.

**Weaknesses:**

- Experiments are limited to GPT2-XL and GPT-J. Evaluation on more modern and widely used LLMs such as Llama or Qwen would make the results more convincing.
- In the ablation study (Table 3), the performance drop from removing each module is relatively small, while the total improvement of AlphaEdit+ over AlphaEdit on AlphaSet is much larger. It would help to analyze which component contributes most to the overall gain.

**Questions:**

- Given that the smoothing iterations and threshold significantly affect both editing performance and efficiency, how should these hyperparameters be selected to balance optimal performance and reasonable runtime?

---

### Note · Authors · 2025-11-24

**Comment:**

We are sincerely grateful to the reviewers for the time and thoughtful effort they dedicated to evaluating our manuscript. After carefully considering all of the comments and suggestions, we have decided to withdraw our current submission to allow for further substantial improvement of the work. The feedback provided is immensely valuable to us, and we will incorporate it closely as we revise and extend this research.

**Withdrawal Confirmation:**

I have read and agree with the venue's withdrawal policy on behalf of myself and my co-authors.